# Aerosol chemistry and particle growth events at an urban downwind site in North China Plain

Yingjie Zhang[1,2,#], Wei Du[1,3,#], Yuying Wang[4], Qingqing Wang[1], Haofei Wang[5], Haitao Zheng[1], Fang Zhang[4], Hongrong Shi[6], Yuxuan Bian[7], Yongxiang Han[2], Pingqing Fu[1], Francesco Canonaco[8], André S. H. Prévôt[8], Tong Zhu[9], Pucai Wang[6], Zhanqing Li[4], and Yele Sun[1,2,3,10]

[1]State Key Laboratory of Atmospheric Boundary Layer Physics and Atmospheric Chemistry, Institute of Atmospheric Physics, Chinese Academy of Sciences, Beijing 100029, China

[2]Collaborative Innovation Center on Forecast and Evaluation of Meteorological Disasters, Nanjing University of Information Science & Technology, Nanjing 210044, China

[3]College of Earth Sciences, University of Chinese Academy of Sciences, Beijing 100049, China

[4]College of Global Change and Earth System Science, Beijing Normal University, Beijing 100875, China

[5]Key Laboratory of 3D Information Acquisition and Application of Ministry of Education, Capital Normal University, Beijing 100048, China

[6]Key Laboratory of Middle Atmosphere and Global Environment Observation, Institute of Atmospheric Physics, Chinese Academy of Sciences, Beijing 100029, China

[7]State Key Laboratory of Severe Weather, Chinese Academy of Meteorological Sciences, Beijing 100081, China

[8]Laboratory of Atmospheric Chemistry, Paul Scherrer Institute, Villigen PSI 5232, Switzerland

[9]College of Environmental Sciences and Engineering, Peking University, Beijing 100871, China

[10]Center for Excellence in Regional Atmospheric Environment, Institute of Urban Environment, Chinese Academy of Sciences, Xiamen 361021, China

[#]These authors contributed equally to this work.

*Correspondence*: Yele Sun (sunyele@mail.iap.ac.cn)

**Abstract.** The North China Plain (NCP) has experienced frequent severe haze pollution events in recent years. While extensive measurements have been made in megacities, aerosol sources, processes, and particle growth at urban downwind sites remain less understood. Here, an aerosol chemical speciation monitor and a scanning mobility particle sizer, along with a suite of collocated instruments, were deployed at the downwind site of Xingtai, a highly polluted city in the NCP, for real-time measurements of submicron aerosol ($PM_1$) species and particle number size distributions during May and June 2016. The average mass concentration of $PM_1$ was 30.5 ($\pm$ 19.4) µg m$^{-3}$, which is significantly lower than that during wintertime. Organic

aerosols (OA) constituted the major fraction of PM$_1$ (38%) followed by sulfate (25%) and nitrate (14%). Positive matrix factorization with the multilinear engine version 2 showed that oxygenated OA (OOA) was the dominant species in OA throughout the study, on average accounting for 78% of OA, while traffic and cooking emissions both accounted for 11% of OA. Our results highlight that aerosol particles at the urban downwind site were highly aged and mainly from secondary formation. However, the diurnal cycle also illustrated the substantial influence of urban emissions on downwind sites, which are characterized by similar pronounced early morning peaks for most aerosol species. New particle formation and growth events were also frequently observed (58% of the time) on both clean and polluted days. Particle growth rates varied from 1.2 to 4.9 nm h$^{-1}$ and our results showed that sulfate and OOA played important roles in particle growth during clean periods, while OOA was more important than sulfate during polluted events. Further analyses showed that particle growth rates have no clear dependence on air mass trajectories.

## 1 Introduction

Atmospheric aerosols can reduce visibility (Zhang et al., 2010), exert adverse effects on human health (Chen et al., 2013), and also affect radiative forcing directly by absorbing and scattering solar radiation and indirectly by modifying cloud formation and properties (Boucher et al., 2013). According to the latest report on global urban air quality by the World Health Organization (WHO), the top 10 most polluted cities in China are all located on the North China Plain (NCP). The concentration of particulate matter (PM) less than or equal to 2.5 μm in diameter (PM$_{2.5}$) in Xingtai ranked the first in China in 2014 (http://apps.who.int/gho/data/view.main.AMBIENTCITY2016?lang=en). Although extensive studies have characterized the formation mechanisms and evolution processes of haze in the NCP (Quan et al., 2011; Zhao et al., 2013; Yang et al., 2015; Li et al., 2017), high anthropogenic emissions and stagnant meteorological conditions are the major factors leading to severe PM pollution (Zhang et al., 2015a; Fu and Chen, 2017; Guo et al., 2014; Sun et al., 2014). Mitigating air pollution in the NCP remains a challenge. One reason is the complexity of ambient aerosols, which have largely different compositions and that come from different sources from different regions and cities.

Xingtai, one of the most polluted cities in China, had an urgent front-burner environmental problem. PM sources are dynamic and include local emissions, e.g., biomass burning, traffic, and cooking emissions, and the transport of pollutants from upwind (east and south) polluted areas (Fu et al., 2014). As a result, both local and regional sources contribute to high concentrations of PM, leading to certain uncertainties in air quality control. Although the annual average concentration of PM$_{2.5}$ in Xingtai decreased from 160 μg m$^{-3}$ in 2013 to 87 μg m$^{-3}$ in 2016 (the data are from four monitoring sites in urban Xingtai that was released by the China National Environmental Monitoring Centre), it still far exceeds the Chinese National Air Quality Standard (35 μg m$^{-3}$ for an annual average) and that of the WHO (10 μg m$^{-3}$). In addition, concentrations of NO$_2$ and CO changed little from 2013–2016 while that of SO$_2$ decreased substantially (Fig. S1). As a response to the changes in precursors,

aerosol particle composition may also change significantly. Therefore, characterization of the composition, sources, and processes of PM in regions near Xingtai is important to do so that effective strategies for future air quality improvements can be provided. Previous studies carried out in Xingtai have investigated the frequency of haze events (Fu et al., 2014), ammonia emissions (Zhou et al., 2015), and the sources of $PM_{2.5}$ (Wang et al., 2015a). The results highlighted the importance of both local (especially industrial) and regional sources of air pollution in Xingtai. However, real-time characterizations of aerosol composition and particle number size distributions have not yet been reported. A recent study conducted in a similarly polluted city, Handan, which is approximately ~50 km south of Xingtai, showed significant contributions of coal and biomass combustion to haze formation in winter (Li et al., 2017). However, aerosol characteristics in Xingtai are not well known and the impacts of urban emissions on downwind sites also remain poorly understood.

In this study, an Aerodyne aerosol chemical speciation monitor (ACSM) along with a suite of collocated instruments was deployed at the downwind site of Xingtai from 30 April to 20 June 2016 to characterize aerosol chemistry and particle growth events in spring and summer. The mass concentrations, chemical composition, and temporal and diurnal variations of submicron aerosol ($PM_1$) species are characterized, and the sources of organic aerosols (OA) are investigated with positive matrix factorization (PMF) and a FLEXible PARTicle dispersion model (FLEXPART) analysis. Also, particle growth events and their relationship to aerosol chemistry are discussed.

## 2 Experimental methods

### 2.1 Sampling site

Xingtai is located in the central-south part of the Beijing-Tianjin-Hebei region with the Taihang Mountains to the west (Fig. 1a). In this work, all measurements were made at the Xingtai National Meteorological Basic Station (XNMBS), a suburban site located approximately 17 km northwest of Xingtai City (37.18°N, 114.37°E; 180 m. a. s. l.), from 30 April to 20 June 2016. The sampling site was influenced by mountain-plain winds during the study period. As shown in Fig. 1b, the wind direction showed clear day and night patterns with prevailing south-southeasterly winds in the day and west-northwesterly winds at night. The average wind speed was 4 m s$^{-1}$ and the average temperature was 22.6 °C during the study period.

### 2.2 Measurements

All instruments were placed in a container at the sampling site. The non-refractory $PM_1$ (NR-$PM_1$) chemical components including sulfate ($SO_4$), nitrate ($NO_3$), ammonium ($NH_4$), chloride (Chl), and organics (Org) were measured in situ by an ACSM at a time resolution of 5 min. The ACSM operated in the same way as in previous studies (Sun et al., 2016b; Zhang et

al., 2016). A PM$_{2.5}$ cyclone (Model: URG-2000-30ED) was supplied in front of the sampling inlet to remove coarse particles larger than 2.5 μm. The ambient air was drawn into the container through a 1/2 inch (outer diameter) stainless steel tube at a flow rate of 3 L min$^{-1}$ using an external pump, of which ~0.1 L min$^{-1}$ was sub-sampled into the ACSM. The sampling height was approximately 2 m, which was 1.5 m higher than the roof of container. Thus, the particle residence time in the sampling tube was about 5 s. Aerosol particles were then dried by a silica gel diffusion dryer before sampling into the ACSM. Before the campaign, the ACSM was calibrated with pure ammonium nitrate particles following the standard protocols in Ng et al. (2011b). Because the ACSM does not measure refractory black carbon (BC), a seven-wavelength aethalometer (model AE33, Magee Scientific Corp.; Drinovec et al., 2015) was used to measure BC.

The size-resolved particle number concentration in the size range from 15 to 685 nm was measured in situ by a condensation particle counter (CPC model 3775, TSI) equipped with a long differential mobility analyser (DMA, model 3081A, TSI). The time resolution is 5 min. The total number concentrations (7–2000 nm) were measured by a Mixing Condensation Particle Counter (MCPC, model 1720, Brechtel). Other collocated measurements included the light extinction of dry PM$_{2.5}$ at 630 nm measured by a Cavity Attenuated Phase Shift Extinction Monitor (CAPS PM$_{ext}$; Massoli et al., 2010), the mass concentration of PM$_{2.5}$ measured by a 6-channel particle counter (manufactured by Graywolf) and gaseous species of CO, NO, NO$_x$, SO$_2$, and O$_3$ measured by gas analyzers (manufactured by ECOTECH). Meteorological parameters including ambient temperature (T), relative humidity (RH), wind speed (WS), wind direction (WD), precipitation, and solar radiation were also measured at the same site by the Xingtai Meteorological Administration.

## 2.3 Data analysis

### 2.3.1 ACSM data analysis

ACSM data were analyzed using the ACSM standard software (version 1.5.3.0) within the Igor Pro software environment (Wave Metrics, Inc., Oregon, USA). The default relative ionization efficiencies for all species except NH$_4$ were used in the study. That for NH$_4$ was determined from the ionization efficiency calibration. A collection efficiency (CE) of 0.5 was used to account for the incomplete detection of aerosol species, mainly due to particle bounce at the vaporizer (Matthew et al., 2008). The CE can be composition dependent and especially sensitive to the fraction of ammonium nitrate (NH$_4$NO$_3$) as well as affected by particle acidity and RH (Middlebrook et al., 2012). In this study, SO$_4$ dominated inorganic species and the average contribution of NH$_4$NO$_3$ was 18% (maximum: 38%), which would not affect CE substantially. Aerosol particles were slightly acidic as indicated by the average ratio (~0.7) of measured NH$_4$ to predicted NH$_4$ that is required to fully neutralize SO$_4$, NO$_3$, and Chl (Zhang et al., 2007), which are also not acidic enough to affect the CE. Default relative ionization efficiencies (RIE) were used except ammonium (5.0) and sulphate (0.98) that were determined from pure ammonium nitrate and ammonium

sulphate, respectively. In addition, to reduce the influence of RH on CE, a silica gel diffusion dryer was deployed to keep the RH in the sampling line below 40%. In fact, the differences in mass concentrations were less than 5% between composition-dependent CE and a constant CE of 0.5 in this study.

Figure S2 shows the comparison between the total $PM_1$ mass (equal to NR-$PM_1$ + BC) and particle volume concentrations measured by the SMPS. Particle volume concentrations were highly correlated with total $PM_1$ mass concentrations ($r^2 = 0.77$, slope = 0.51). We then estimated the particle density using the chemical composition of $PM_1$ (Salcedo et al., 2006). The average density during the study period was 1.5 g $cm^{-3}$. Assuming spherical particles, the calculated SMPS mass reports 75% of the total $PM_1$ mass. Such a difference may be caused by measurement uncertainties between different instruments, the effects of particle shape and the uncertainties in estimating particle density.

### 2.3.2 Positive matrix factorization (PMF) analysis

To determine the sources of OA, ACSM mass spectra were processed using the Multilinear Engine version 2 (ME-2) algorithm implemented with the toolkit called Source Finder (Canonaco et al., 2013). The so-called *a*-value approach allows the user to introduce a priori information in the form of known factor profiles or time series to obtain a unique solution and thus reduce the rotational ambiguity of the PMF algorithm. The mass spectra and error matrices of OA were prepared according to the procedures detailed by Ulbrich et al. (2009) and Zhang et al. (2011). Given the interference of the internal standard of naphthalene at *m/z* 127–129 and the low signal-to-noise ratios of larger ions, we only considered *m/z* values below 120 in this study. A reference hydrocarbon-like OA (HOA) profile, which is the average of multiple ambient data sets (Ng et al., 2011a), and a reference cooking OA (COA) profile in Beijing (Sun et al., 2013) were introduced to constrain the model performance with *a*-values varying from 0 to 1. Following the guidelines presented by Canonaco et al. (2013) and Crippa et al. (2014), an optimal solution involving three factors with an *a*-value of 0.2 was accepted. Some important criteria for selecting the optimal solution with *a*-value varying from 0 to 1 are shown in Figs. S3-S7. The mass spectra and time series of three OA factors are shown in Fig. 2.

The HOA factor has a similar mass spectrum as that of freshly emitted traffic or other fossil-fuel combustion aerosols with major peaks at *m/z* equal to 41, 43, 55, and 57. HOA was moderately correlated with BC ($r^2 = 0.42$). The COA resolved in Xingtai had an *f*55 to *f*57 ratio of 2.3, within the range of values for COA ($\sim > 1.2$) (Mohr et al., 2011). We further evaluated the factors of HOA and COA assuming that BC is dominantly from traffic emissions while the contribution from cooking emissions is minor (Sun et al., 2018). POA from the two-factor solution was highly correlated ($r^2 > 0.66$) with BC between 1:00 – 10:00 when cooking emissions were not significant (Fig. S8). The ratios of POA/BC were also the lowest during this period, suggesting the dominant contribution of HOA to POA. We then used the average ratio of POA/BC (0.62) during this

period to estimate the concentrations of HOA and COA. The estimated HOA and COA on average contributed both 11% to OA, consistent with the results of ME-2 analysis. This suggested that the results from ME-2 analysis are reasonable.

The mass spectrum of oxygenated OA (OOA) is characterized by a prominent peak at $m/z$ 44 (23.6% of the total OOA signal), which has also been reported in previous studies. In addition, OOA was highly correlated with sulfate ($r^2 = 0.75$) and moderately correlated with nitrate ($r^2 = 0.54$), suggesting that OOA is a surrogate of secondary OA (SOA; Fig. 2). We also performed a PMF analysis by applying the PMF2 algorithm to the ACSM-measured OA. Although the two-factor solution identifies a primary OA (POA) and an OOA, solutions with three to five factors show a splitting and mixing of factors. Therefore, the ME-2 algorithm was used in this study.

**2.4 Source region analysis**

The footprints of the selected episodes were determined using backward simulations from FLEXPART, a Lagrangian transport and dispersion model (Stohl et al., 2005). The model calculated the 36-h backward trajectories of 10,000 particles released every hour from the sampling site at a height of 180 m above sea level. The meteorological data driving the model were simulated by version 3.4 of the Weather Research and Forecasting (WRF) model with a1-h time resolution and a 10-km spatial resolution. The WRF model was driven by initial and boundary conditions from National Centers for Environmental Prediction global reanalysis data.

Seventy-two-hour back trajectories at a height of 500 m were calculated every hour using the Hybrid Single-Particle Lagrangian Integrated Trajectory (HYSPLIT) model (Stein et al., 2015) at the XNMBS. To investigate the chemical characteristics of aerosols from different source regions, a cluster analysis was then performed on the trajectories and three clusters were identified according to their similarities in spatial distributions.

In addition, non-parametric wind regression (NWR; Petit et al., 2017) was performed to evaluate the sources of local emissions and regional transport for $PM_1$ aerosol species and OA factors. The NWR plots represent the probability that a specific compound or source is located in a certain wind direction.

## 3 Results and discussion

### 3.1 Aerosol composition and temporal variations

The temporal variations of $PM_1$ aerosol species and meteorological parameters (RH, $T$, WS, WD, and precipitation) are shown in Fig. 3. The average mass concentration of $PM_1$ (equal to NR-$PM_1$ + BC) was 30.5 μg m$^{-3}$ and ranged from 0.2 to 140.1 μg m$^{-3}$. Several pollution episodes usually lasting ~2–3 days were observed during the study period, e.g., on 9-11, 17–23, 28–31 May, and 2–4 June. These pollution episodes were quickly cleaned mainly by wet scavenging. The temporal variations varied differently among different chemical species. Organics showed dramatic variations, ranging from 0.01 to 101.5 μg m$^{-3}$, and comprised the major fraction of $PM_1$ for most of the time in this study. High concentration peaks of organics were frequently observed, likely due to the influences of local emissions. By contrast, sulfate concentrations obviously increased and remained relatively high during the pollution events, suggesting the important role of regional transport at the downwind site of Xingtai. The average $PM_{2.5}$ mass concentration was 45.2 μg m$^{-3}$. Although the average $PM_{2.5}$ mass concentration was 15% lower than that (53.3 μg m$^{-3}$) measured at the urban sites in Xingtai, it exceeded the Chinese National Ambient Air Quality Standards by 29%. These results suggest that the urban downwind sites also experiences similar PM pollution events as the urban sites. For example, the daily $PM_{2.5}$ exceeded the Chinese National Ambient Air Quality Standards 24% of the time during this study. $PM_1$ was highly correlated with $PM_{2.5}$ ($r^2 = 0.95$) and on average, comprised ~64% of $PM_{2.5}$. The average mass concentration of $PM_1$ is close to that measured in Xinzhou (35 μg m$^{-3}$), a city in central China (Wang et al., 2016), but lower than that measured in 2013 in Xianghe (73 μg m$^{-3}$), a rural site near Beijing (Sun et al., 2016a; Fig. S9). One possible explanation is the significant improvement in air quality during the last four years (Fig. S1).

On average, OA was the largest component of $PM_1$, accounting for 38% of the total $PM_1$ mass, followed by sulfate (25%), nitrate (14%), ammonium (10%), and BC (10%; Fig. 1a). POA (equal to HOA + COA) and SOA (OOA) accounted for 22% and 78%, respectively, of the total OA mass. Together, ~18% of $PM_1$ was comprised of primary-related materials (POA + BC) and 82% was from secondary formation ($NO_3$ + $SO_4$ + $NH_4$ + Chl + SOA), indicating that aerosol particles from secondary aerosol formation processes dominated at the downwind site of Xingtai. Compared with aerosol composition in megacities in the NCP, e.g., Beijing (Hu et al., 2016), aerosol composition in this study showed substantially higher contributions of SOA (29%) and BC (10%), while the contributions of $NO_3$ (14%) and COA (4%) were low due to the lesser amount of local traffic and cooking emissions. The nitrate contribution was similar to that observed at a suburban site in Xinzhou (Wang et al., 2016), but the $SO_4$ contribution was relatively low (25% versus 32%). One reason is that the higher RH (70% versus 52%) in Xinzhou facilitated the formation of $SO_4$. The $PM_{2.5}$ concentration during this study period was more than twice lower than that in winter (151 μg m$^{-3}$, Fig. S10), suggesting less dilution in winter but also that PM came from different sources in spring-summer and winter. For example, the winter season has significantly enhanced coal combustion emissions.

We also investigated the compositional differences between clean periods and polluted events (Fig. 3). Secondary inorganic aerosols (SIA) including $SO_4$ (26% versus 21%), $NO_3$ (15% versus 9%), and $NH_4$ (10% versus 7%) showed enhanced contributions to $PM_1$ during polluted events, while a corresponding decrease was found for organics. For example, the contribution of SIA increased by 14% from 37% during clean periods to 51% during polluted events, while organics and BC decreased by 9% and 3%, respectively. Although SOA dominated the OA composition during both polluted events and clean periods, SOA also contributed more to OA during polluted events (78% vs 72%). Comparatively, POA (HOA and COA) was relatively more important than SOA during clean periods. These results suggest that PM at the downwind site of Xingtai was mainly affected by regional transport and secondary formation.

### 3.2 Diurnal patterns

Aerosol species showed distinctly different diurnal patterns in this study (Fig. 4), indicating that the sources and formation processes of PM pollutants were different. The diurnal cycle of $PM_1$ was characterized by peaks at ~1000 and ~2100 local time (LT). The first peak in the late morning was consistent with that of gas pollutants including CO, $SO_2$, and $NO_x$ (Fig. S12). Through comparisons with the diurnal cycle of $PM_{2.5}$ at the urban site in Xingtai (Fig. S13) and daily variations in WD (Fig. 1), the morning peak was mainly associated with the transport of pollutants from urban sites located to the southeast.

The high concentration of $PM_1$ at night was clearly associated with enhanced primary emission concentrations, e.g., HOA and COA, although the shallower boundary layer may have also played a role. It is interesting to note that $b_{ext}$ did not show a pronounced nighttime peak as did $PM_{2.5}$, indicating that the extinction coefficients of primary aerosols were smaller than those of secondary aerosol species (Wang et al., 2015b). The diurnal pattern of $PM_1$ only showed a pronounced peak at night during clean periods, consistent with those of HOA and COA. While the diurnal variations support the enhanced roles of primary emissions for PM during clean periods, they also indicate the lesser influences from urban emissions during daytime.

The diurnal pattern of organics overall resembles that of $PM_1$ and was characterized by two pronounced peaks between 0900–1200 LT and 1900–2100 LT. However, the three OA factors showed different diurnal cycles. The average diurnal cycle of HOA showed a small morning peak and a pronounced nighttime peak. While the two peaks were comparable during polluted periods, only the pronounced nighttime peak was observed during clean periods. As shown in Fig. S11a, winds were dominantly from the south-southeast during polluted periods and mainly from the west-northwest during clean periods. These results suggest that the high morning HOA peak was mainly caused by the transport of pollutants from urban sites, while local traffic emissions became an important source of HOA at night. This is also consistent with what is seen in the corresponding bivariate polar plot (Fig. 5h), which shows high concentrations of HOA in regions to the south-southeast and to the north. The decrease in HOA during the day was mainly associated with the rising planetary boundary layer height.

The diurnal pattern of COA (Fig. 4h) was also similar to that observed in megacities (Sun et al., 2013; Crippa et al., 2013; Elser et al., 2015; Zhang et al., 2015b; Hu et al., 2016; Xu et al., 2016) and was characterized by two peaks around the mealtime hours, reflecting the influence of cooking emissions. However, the COA concentration in this study was much lower than that reported in megacities (e.g., peak concentration: 2.5 μg m$^{-3}$ versus 14 μg m$^{-3}$ in Beijing; Hu et al., 2016), suggesting much less cooking emissions at the suburban site. The corresponding bivariate polar plot further shows that the high concentration of COA mainly originated from restaurants and inhabitants' activities to the southeast of the site (Fig. 5i).

Overall, the diurnal cycle of OOA (Fig. 4i) was flat during both clean and polluted periods in this study, reflecting regional characteristics. This is also consistent with the wide distribution of OOA seen in the corresponding bivariate polar plot (Fig. 5j). Secondary inorganic species of nitrate and sulfate had different diurnal profiles. Nitrate had a pronounced diurnal cycle with much higher concentrations at night than during the day. As shown in Fig. 4d, the nitrate concentration decreased from 6.0 μg m$^{-3}$ to 2.6 μg m$^{-3}$ between 1100 LT and 1900 LT, which was mainly due to the evaporative loss of particulate ammonium nitrate particles due to high temperatures. Such diurnal cycles have been observed many times during the summer in megacities, e.g., Beijing (Sun et al., 2012), Nanjing (Ge et al., 2017), Lanzhou (Xu et al., 2016), and also on other continents (e.g., Lanz et al., 2007). Nitrate also showed an increase in the early morning after sunrise (Fig. 4d). While the transport from urban sites played a role, this increase was mainly caused by daytime photochemical production when $T$ was not high enough to substantially affect gas partitioning. During daytime clean periods, the diurnal cycle of nitrate was flat with higher concentrations at night. Sulfate showed a much smoother diurnal evolution compared with nitrate (Fig. 4c), reflecting the regional characteristics of sulfate. Two small peaks were observed during the day. While the first peak between 0900–1100 LT was most likely from urban transport, consistent with regional sources from the southeast indicated by the corresponding bivariate polar plot (Fig. 5c), the second one was more likely from daytime photochemical production. Chloride accounted for a small fraction of the PM$_1$ mass, yet it showed a strong diurnal cycle with a pronounced peak in the morning (Fig. 4e). This peak was noticeably similar to those of CO and SO$_2$ (Figs. S11a and S11c), suggesting that the dominant source was from combustion emissions in the southeast.

**3.3 Particle number size distributions**

Figure 6 shows the time series of total number concentration (15–685 nm, $N_{15-685}$) and three different modes including the small Aitken mode (15–40 nm, $N_{15-40}$), the large Aitken mode (40–100 nm, $N_{40-100}$), and the accumulation mode (100–400 nm, $N_{100-685}$), as well as particles in the range of 7–15 nm ($N_{7-15}$) that was calculated from the difference between MCPC and SMPS measurements. The corrections for diffusion loss and multiple charge have been applied in SMPS data analysis. The average total number concentration was 11,200 ± 5800 cm$^{-3}$, which is comparable to that measured in Shangdianzi (12,000 cm$^{-3}$; Shen et al., 2011), and Yufa (10200 cm$^{-3}$; Peng et al., 2014), and slightly higher than that observed in Beijing (10100 cm$^{-3}$; Du et

al., 2017). $N_{15-685}$ showed a pronounced diurnal cycle with a clear increase during the day (Fig. 7e). A further analysis highlights that this increase was mainly driven by small and large Aitken mode particles, indicating the impacts of new particle formation and growth on the diurnal variations in particle number (Fig. S14). For example, the small Aitken-mode particles and ultrafine particles showed rapid daytime increases after sunrise by more than a factor of 5 during both polluted events and clean periods. Note that the number concentrations of $N_{7-15}$ during clean periods was much higher than during polluted events (6600 cm$^{-3}$ versus 3300 cm$^{-3}$, on average), which is comparable to the number concentration of small Aitken-mode particles. These results show that (1) new particle formation was much stronger on clean days than on polluted days, and (2) new particle formation also occurred on polluted days with newly-formed particles growing quickly due to the higher condensation sink (CS; 0.05 s$^{-1}$). About 49% of the total particle number was made up of large Aitken-mode particles. However, these particles accounted for a small fraction of the total volume concentration (5%). We also note that the pronounced nighttime peak in $N_{40-100}$ coincidently agrees with that of COA during clean periods, suggesting the influence of cooking emissions on large Aitken-mode particles. We calculated the particle number size distribution for two nights that experienced significant cooking emission events, i.e., 2 May and 13 May. Narrow single-mode distributions peaking at ~60 nm were seen (Fig. S15), supporting the influence of cooking emissions on large Aitken-mode particles. The diurnal cycle of $N_{100-685}$ was relatively flat except for a small morning peak (Fig. 7d). This is also consistent with results from the corresponding bivariate polar plot (Fig. S16d). The number concentration of $N_{100-685}$ during polluted events was more than a factor of 3–4 times that during clean periods, which contributes toward the major difference in particle number characteristics between polluted events and clean periods. One reason is the higher CS during polluted events, which facilitated the growth of particles.

Figure 8 shows that the average particle number concentration had a bimodal size distribution with the geometric mean diameter (GMD) peaking at 46 nm and 106 nm, respectively. Note that the peak diameter for the entire study was ~62 nm, which is higher than that observed in urban Beijing (45 nm; Du et al., 2017). The higher CS (0.36 s$^{-1}$ versus 0.29 s$^{-1}$) could be one of the reasons leading to more water vapor condensing on preexisting particles at the rural site. This is also consistent with the shift in the peak diameter from 57 nm during clean periods to 88 nm during polluted events. Although similar bimodal size distribution modes peaking at ~46 nm (41 nm) and ~109 nm (106 nm) were observed during clean periods and polluted events, respectively, the relative contributions of the two modes were largely different. While the particle number concentration was dominated by the small mode (56%) during clean periods, the large mode was more important during polluted events, accounting for 73% of the particle number concentration (Table 1). These results confirm the different roles of different mode particles between clean periods and polluted events.

### 3.4 Particle growth events

New particle growth events (NPE) were frequently observed during the study period. As shown in Fig. 9a, particle growth

typically started at ~0800 LT and ended at midnight with an increase in GMD from ~25 nm to ~60 nm. This result is consistent with those previously reported for rural sites in the NCP (Wang et al., 2013) and urban sites, e.g., Beijing (Du et al., 2017). Note that the growth sizes of particles were overall larger than those observed in urban Beijing (from ~22 nm to ~55 nm), likely indicating a stronger aging process at the suburban site. The growth of particles tracked the diurnal cycle of CS, which showed a continuous increase from early morning to midnight. Although ACSM has a good transmission for only particles within the size range between 70 and 500 nm in diameter(Jayne et al., 2000), some particles from the small Aitken mode might not be detected by the ACSM. Simultaneous comparison between aerosol chemical composition and particle size distribution during the growth period make some sense. Aerosol composition seemed to significantly change during the growth period. As shown in Fig. 9a, OOA and sulfate were the only two species whose contributions increased, going from 26% to 33% and 27% to 33%, respectively, during the growth period (1000–1800 LT). Although the increases in sulfate and OOA were partly due to the decreases in nitrate and chloride because of the evaporative loss in the afternoon, the CO-normalized sulfate and OOA ($SO_4$/CO and OOA/CO) also showed increases. Note that CO here was subtracted by a background value of 0.068 ppm that was calculated as the average of the lowest 5% data in this study. These results show that both sulfate and OOA played important roles in daytime particle growth.

We also examined particle growth events on polluted days and clean days. As shown in Figs. 9b and 9c, the growth process of particles on polluted days started at ~1200 LT with the GMD increasing from ~38 to ~57 nm in six hours. Particle growth started earlier (~1000 LT) on clean days with the GMD increasing from ~27 to ~41 nm. Particle growth on polluted days was faster than on clean days, which was likely due to the higher CS on polluted days. This is also consistent with more significant increases in SOA/CO and sulfate/CO on polluted days. The increase in CS between 0900–1200 LT on polluted days was associated with the corresponding increases in most aerosol species. However, the CO-normalized aerosol species did not show such an increase during this period suggesting that aerosol species and gaseous species were from the same air mass, i.e., transported in from urban sites. Such a pattern was not observed on clean days. Particle growth at night was more clearly seen on clean days than on polluted days, consistent with the increase in OOA/CO. However, such an increase was not observed for sulfate/CO on clean days, suggesting that OOA played a more important role than sulfate in particle growth at night.

We further calculated the particle growth rate (GR) of each growth event that lasted more than three hours (Fig. 10a and Table S1). The particle growth rates were calculated using Eq. (1).

$$GR=\Delta D_m/\Delta t \qquad\qquad\qquad (1)$$

Where $D_m$ is the geometric mean diameter from the log-normal fitting of each size distribution and $\Delta D_m$ is the increase in

diameter during the growth period of $\Delta t$. The particle GR varied from 1.2 nm h$^{-1}$ to 4.7 nm h$^{-1}$, which generally falls within the range of values (1-20 nm h$^{-1}$) based on observations from around the world (Kulmala et al., 2004; Yu et al., 2017). As indicated in Fig. 10b, GR was positively correlated with CS for most of the time, mainly for the fraction of OOA higher than 30% ($r^2 = 0.61$, Fig. S17a). As CS increased from ~0.01 s$^{-1}$ to 0.05 s$^{-1}$, GR increased from 3 nm h$^{-1}$ to 5 nm h$^{-1}$. Higher CS is usually associated with lower particle GR due to faster consumption of condensable vapours. The positive relationship between GR and CS might indicate that the source of condensable vapours contributes to the increase in CS prior to the observation. Another possible explanation is that heterogeneous surface chemistry was more important than CS in growing aerosol particles in highly polluted environment (Kulmala et al., 2017). Figure 10b also showed some low GRs with high CS, which were characterized by low contributions of OOA. These results supports the importance of the involvement of OOA in particle growth (Wu et al., 2017). Figure 10c further shows that GR was positively correlated with the concentration of sulfate during periods with low sulfate mass loadings (< 3 μg m$^{-3}$, $r^2$=0.42, Fig. S17b), while periods with higher concentrations of sulfate had lower GRs. This is corroborated by Fig. S18, showing that GR decreased as the sulfate contribution increased. By contrast, GR was positively correlated with OOA/PM$_1$ for most of the time. These results highlight that OOA played a more important role in particle growth than sulfate although sulfate was also important during periods with low mass loadings. We further checked the dependence of GR on source region, but no clear relationship was found. For example, GR varied from 1.6–3.2 nm h$^{-1}$ during periods with air masses from the north, 1.8–4.9 nm h$^{-1}$ from the southwest, and 1.8–4.6 nm h$^{-1}$ from the east-southeast (Fig. 11). This shows that GR had no clear dependence on air masses from different regions although sulfate concentrations showed much difference.

**3.5 Aerosol composition and particle number concentrations from different source regions**

Figure 12 shows the average composition and particle number distributions from different source regions. The air masses in cluster 3 (C3, 39% of the time) were mainly from the southeast while the other two clusters were mainly from the northwest. The average PM$_1$ concentration for C3 was 42 μg m$^{-3}$, which is ~ 30% higher than that from the other two source regions. The high mass loadings for this cluster suggest that source regions southeast of the site were responsible for the high PM pollution at the sampling site. This is also supported by the bivariate polar plots of aerosol species with high concentration regions in the southeast (Fig. 5). Among all clusters, the two dominant species in PM$_1$ were SOA (26–28%) and sulfate (26–29%). Overall, the aerosol bulk composition for the three clusters from different source areas was similar. These results suggest that aerosol particles were relatively well mixed over the region around Xingtai. Particle number concentrations showed more differences among the different clusters. The two clusters from the northwest were both dominated by large Aitken-mode particles, on average accounting for 49% and 51% of the particles making up the clusters, respectively. Although large Aitken-mode particles dominated the total particle number for C3, we also observed large increases in accumulation-mode particles (39%) compared with the other two clusters (28–30%). These results suggest that air masses from the northwest were relatively clean,

which led to a more frequent occurrence of new particle formation and growth events, while those from the southeast with higher PM loadings tended to form more large particles due to the high CS.

**4 Conclusions**

We presented an analysis of aerosol chemistry and particle growth events at an urban downwind site in the NCP during May and June of 2016 using real-time measurements from an ACSM, an SMPS, and a suite of collocated instruments. Our results showed that the $PM_1$ level in spring and summer (30.5 μg m$^{-3}$) was much lower than that during wintertime when coal combustion emissions were enhanced. Similar to previous studies with a focus on the NCP, aerosol composition at the downwind site of Xingtai was dominated by organics (38%), 78% of which was identified as secondary OA according to the ME-2 analysis. Secondary aerosols (i.e., SNA + SOA) accounted for 78% of $PM_1$, highlighting the major source of secondary formation and regional transport at the downwind site. Local sources of PM from traffic and cooking emissions accounted for less than 10% of the total $PM_1$ mass. We also observed daytime transport from urban sites leading to similar pronounced early morning peaks for most aerosol species. New particle growth events were frequently observed (58% of the time) during the study period. By linking the GR with aerosol composition, we found that OOA and sulfate were two major species affecting the growth of particles. In particular, both OOA and sulfate played important roles in particle growth during clean periods, while OOA was more important than sulfate during polluted events. This is also supported by the decrease in GR as the sulfate contribution increased. A further analysis showed that particle growth rates have no clear dependence on air mass trajectories.

*Data availability.* The data in this study are available from the authors upon request (sunyele@mail.iap.ac.cn).

*Author contributions.* YS and ZL designed research; YZ, WD, YW, QW, HW, HZ, FZ, HS, YB, YH, PF, TZ, PW, and YS performed research; FC and AP shared the data analysis software; YZ, WD, and YS analyzed the data and wrote the paper.

*Competing interests.* The authors declare that they have no conflict of interest.

*Special issue statement.* This article is part of the special issue "Regional transport and transformation of air pollution in eastern China". It is not associated with a conference.

*Acknowledgements.* This study was supported by the National Key Project of Basic Research (2013CB955801, 2014CB447900) and the National Natural Science Foundation of China (41575120).

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

**Table 1.** Comparison of aerosol properties and meteorological parameters between polluted events and clean periods.

| | Polluted Events | Clean Periods |
|---|---|---|
| Mass concentration ($\mu$g m$^{-3}$) | | |
| $SO_4$ | 11.1 | 2.3 |
| $NO_3$ | 6.2 | 1.0 |
| $NH_4$ | 4.4 | 0.8 |
| Chl | 1.1 | 0.4 |
| BC | 4.0 | 1.3 |
| Org | 16.2 | 5.4 |
| HOA | 1.66 | 0.68 |
| COA | 1.67 | 0.73 |
| OOA | 12.2 | 3.6 |
| Particle number concentration (cm$^{-3}$) | | |
| $N_{15-40}$ | 2176 | 2471 |
| $N_{40-100}$ | 6095 | 5308 |
| $N_{100-685}$ | 5424 | 2977 |
| $N_{15-685}$ | 13696 | 10756 |
| Gaseous species | | |
| CO (ppm) | 1.3 | 0.8 |
| $O_3$ (ppb) | 81.2 | 48.9 |
| $NO_x$ (ppb) | 35.0 | 26.2 |
| NO (ppb) | 6.4 | 5.6 |
| $SO_2$ (ppb) | 15.9 | 4.4 |
| Meteorological parameters | | |
| $T$ (°C) | 22.9 | 18.6 |
| RH (%) | 52.7 | 41.3 |

**Figure Captions:**

**Figure 1:** (a) Location of the sampling site. (b) average diurnal evolution of wind vector. The pie chart in (a) shows the average aerosol composition for the entire study. The two arrows in (a) show the daytime and nighttime prevailing wind directions.

**Figure 2:** Mass spectral profiles (on the left) and time series of the mass concentrations of three OA factors (on the right), i.e., HOA, COA, and OOA. Times series of the mass concentrations of BC and sulfate (right axis) are also shown.

**Figure 3:** Time series of (a) temperature ($T$, in black) and relative humidity (RH, in magenta), (b) wind direction (WD, in orange) and wind speed (WS, in black), (c) particle extinction coefficient (Ext., in orange) and precipitation (Precip., in purple), (d) particle number size distribution, (e) mass concentrations of Org, $NO_3$, $SO_4$, $NH_4$, Chl, and BC, and (f) mass fractional contribution of chemical species to total $PM_1$. Polluted events (PE) and clean periods (CP) are marked as shaded orange and blue areas, respectively.

**Figure 4.** Average diurnal cycles of chemical species of $PM_1$ and OA factors during the entire study period, polluted events (PE) and clean periods (CP).

**Figure 5:** Bivariate polar plots of $PM_1$ species as a function of wind speed and wind direction: (a) $PM_1$, (b) Org., (c) $SO_4$, (d) $NO_3$, (e) $NH_4$, (f) Chl, (g) BC, (h) HOA, (i) COA, and (j) OOA. The color scales in (a) – (j) range from 0 to 44, 17, 11, 7.2, 4.5, 1.9, 4.1, 2.0, 2.7, and 12 µg m$^{-3}$, respectively.

**Figure 6:** Time series of particle number concentrations for (a) $N_{7-15}$ calculated from the differences between MCPC and SMPS measurements, (b) $N_{15-40}$ (15–40 nm), (c) $N_{40-100}$ (40–100 nm), (d) $N_{100-685}$ (100–685 nm), and (e) all particles, $N_{15-685}$ (15–685 nm). The gap in (a) is mainly due to the malfunction of MCPC during this period.

**Figure 7:** Diurnal cycles of particle number concentration for (a) $N_{7-15}$ calculated from the differences between MCPC and SMPS measurements, (b) $N_{15-40}$ (15–40 nm), (c) $N_{40-100}$ (40–100 nm), (d) $N_{100-685}$ (100–685 nm), and (e) all particles, $N_{15-685}$ (15–685 nm). Overall mean cycles are shown as black lines. Mean cycles for polluted events (PE) and clear periods (CP) are shown as red and blues lines, respectively.

**Figure 8:** Average particle number size distributions during (a) the entire study, (b) polluted events, and (c) clean periods.

**Figure 9:** Average diurnal evolution of particle number size distributions and aerosol composition for new particle growth events during (a) the entire study, (b) polluted events, and (c) clean periods. The black solid lines in the top three panels show the diurnal cycles of CS. The circles and squares show the GMD from the log-normal fitting from this study and in Beijing (Du et al., 2017), respectively. The average diurnal cycles of aerosol species and CO-normalized aerosol species during polluted events and clean periods are shown in (b) and (c).

**Figure 10:** (a) Particle growth rates (GR) and the corresponding aerosol composition during the growth period, (b) the relationship between GR and CS, color-coded by OOA/$PM_1$, and (c) the relationship between GR and sulfate concentration, color-coded by $SO_4$/$PM_1$. The numbers over the circles in (b) represent different source regions (Fig. 11) and the triangles represent data without information about composition.

**Figure 11:** Particle GR and sulfate concentrations during the days shown in Fig. 10a with different footprints: (A) north, (B) southwest, and (C) east-southeast.

**Figure 12:** Average composition of aerosol particles and particle number concentration for three different clusters.

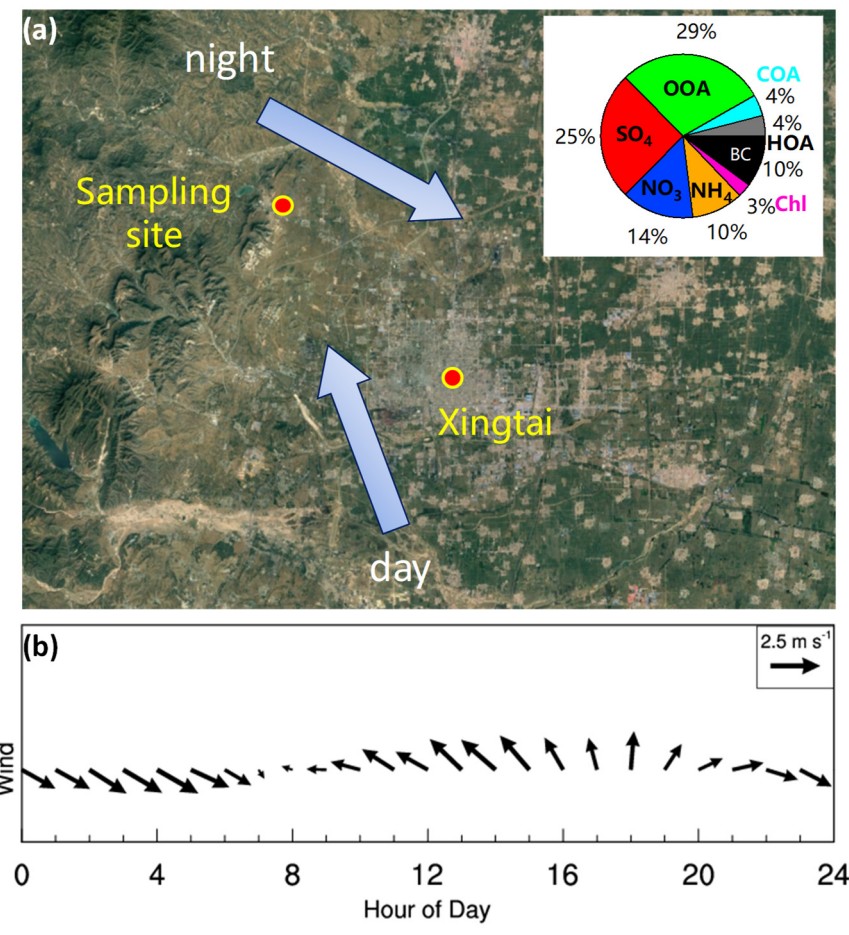

**Figure 1:** (a) Location of the sampling site. (b) average diurnal evolution of wind vector. The pie chart in (a) shows the average aerosol composition for the entire study. The two arrows in (a) show the daytime and nighttime prevailing wind directions.

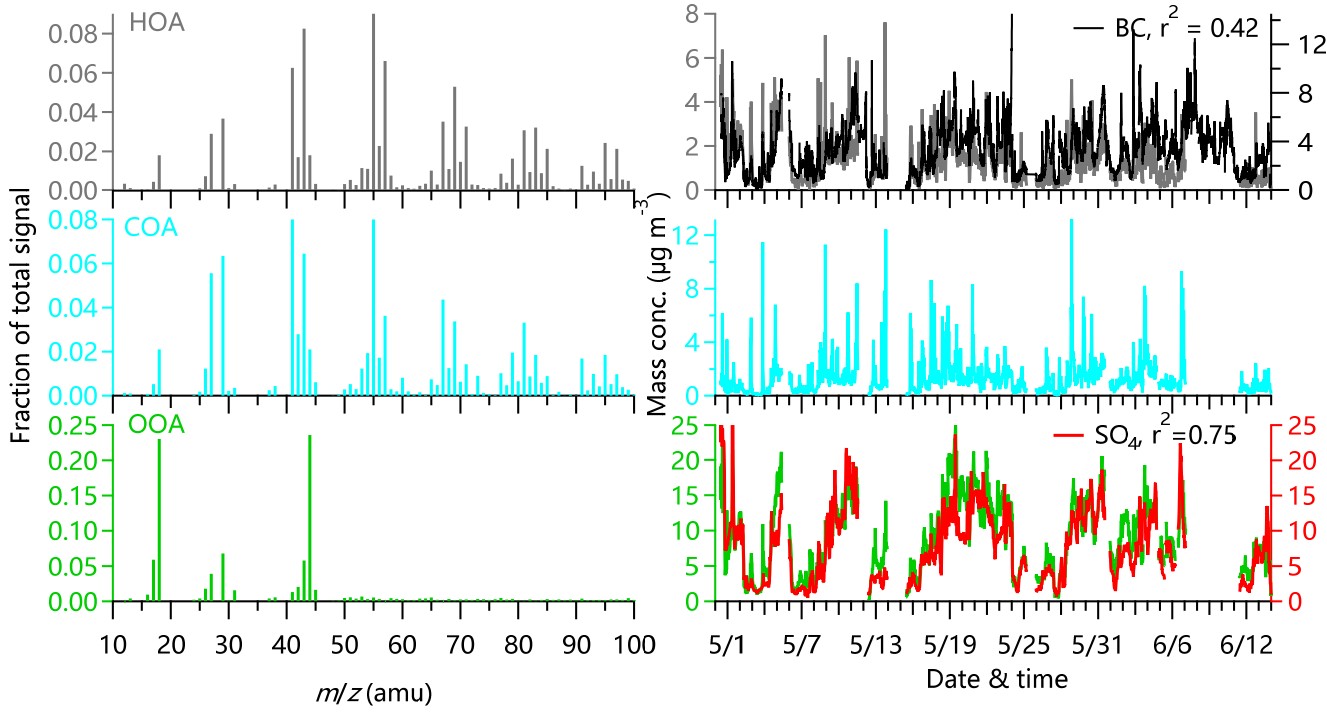

**Figure 2:** Mass spectral profiles (on the left) and time series of the mass concentrations of three OA factors (on the right), i.e., HOA, COA, and OOA. Times series of the mass concentrations of BC and sulfate (right axis) are also shown.

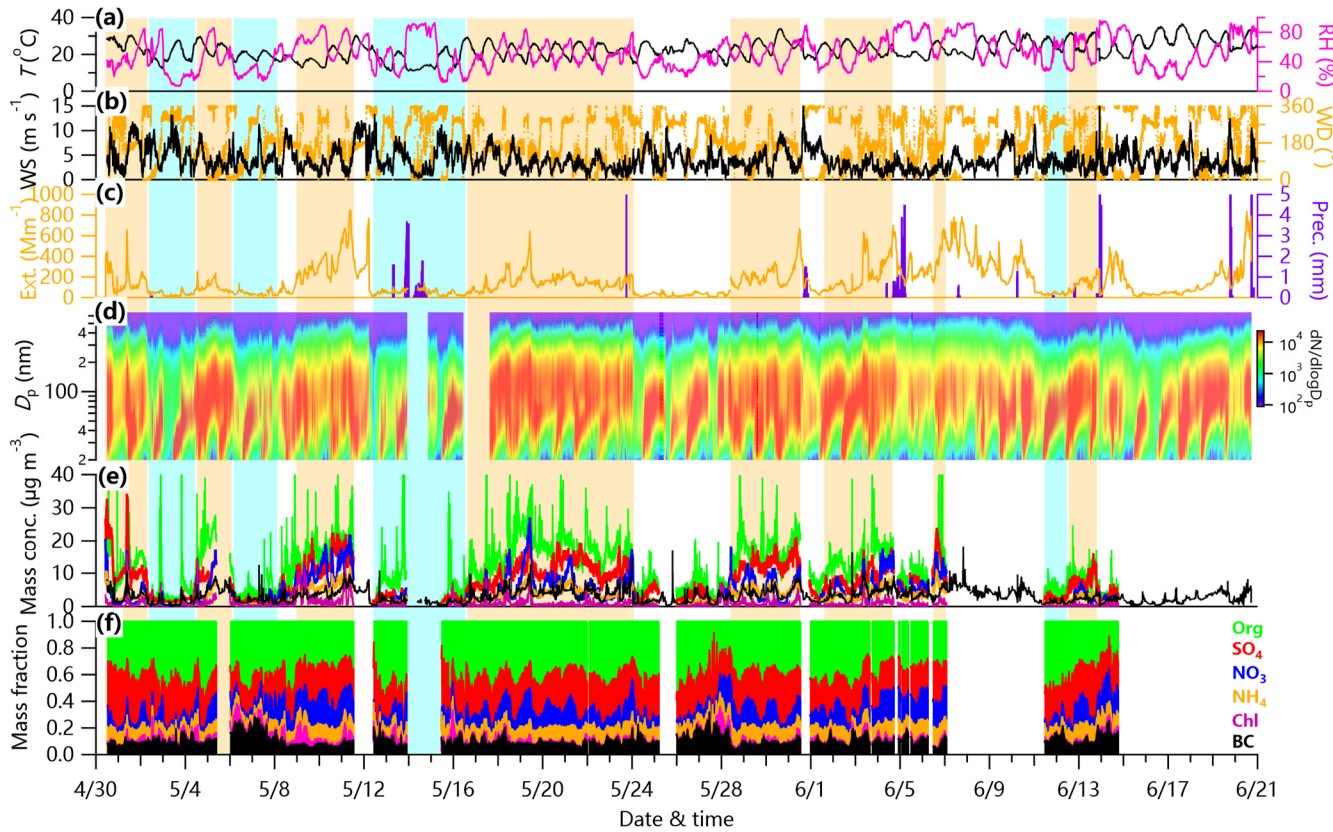

**Figure 3:** Time series of (a) temperature ($T$, in black) and relative humidity (RH, in magenta), (b) wind direction (WD, in orange) and wind speed (WS, in black), (c) particle extinction coefficient (Ext., in orange) and precipitation (Precip., in purple), (d) particle number size distribution, (e) mass concentrations of Org, $NO_3$, $SO_4$, $NH_4$, Chl, and BC, and (f) mass fractional contribution of chemical species to total $PM_1$. Polluted events (PE) and clean periods (CP) are marked as shaded orange and

blue areas, respectively.

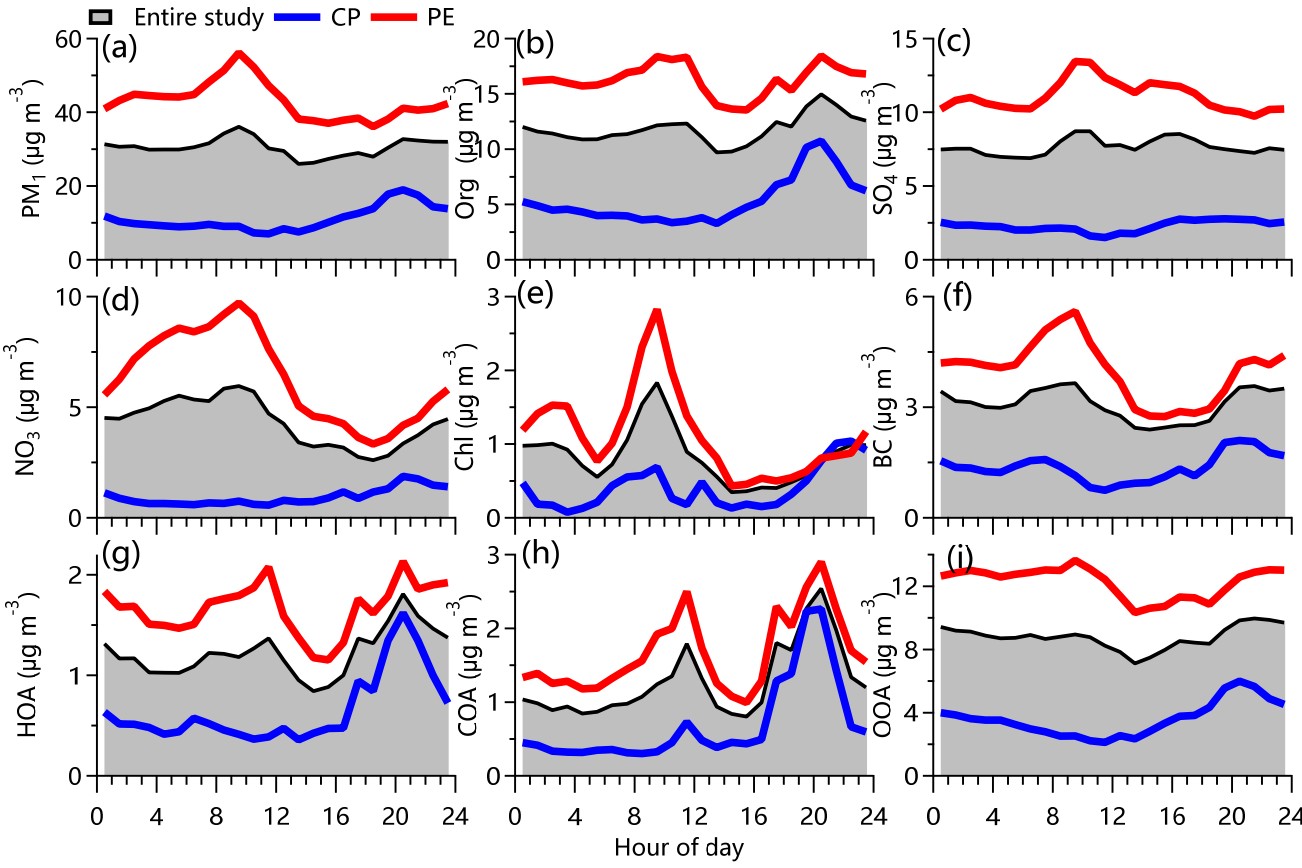

**Figure 4.** Average diurnal cycles of chemical species of PM$_1$ and OA factors during the entire study period, polluted events (PE) and clean periods (CP).

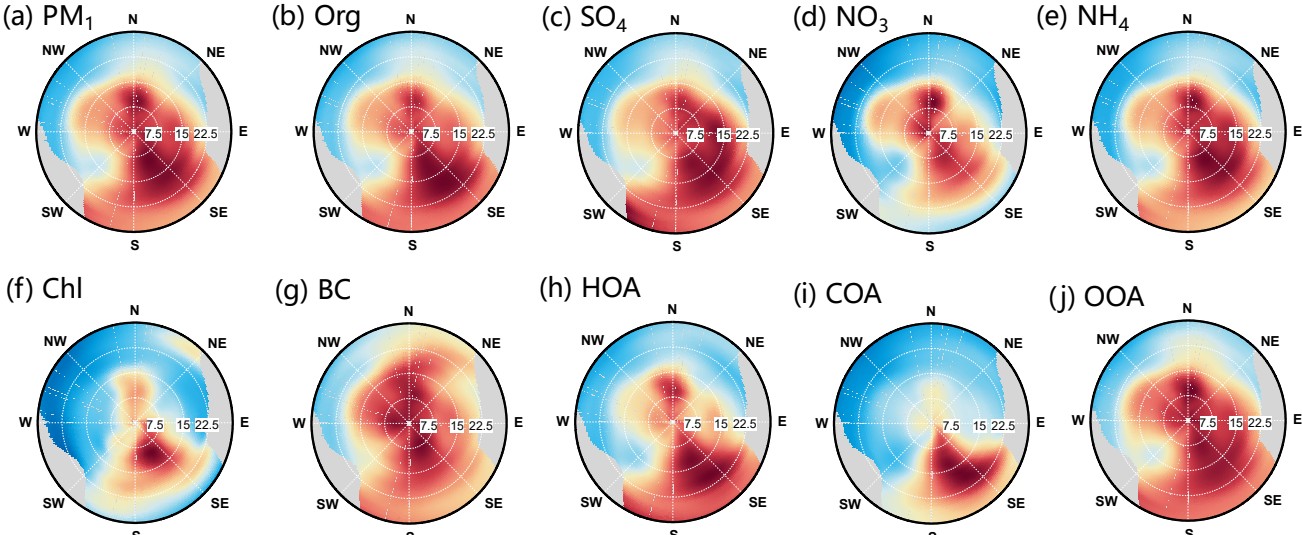

**Figure 5:** Bivariate polar plots of $PM_1$ species as a function of wind speed and wind direction: (a) $PM_1$, (b) Org., (c) $SO_4$, (d) $NO_3$, (e) $NH_4$, (f) Chl, (g) BC, (h) HOA, (i) COA, and (j) OOA. The color scales in (a) – (j) range from 0 to 44, 17, 11, 7.2, 4.5, 1.9, 4.1, 2.0, 2.7, and 12 $\mu g\ m^{-3}$, respectively.

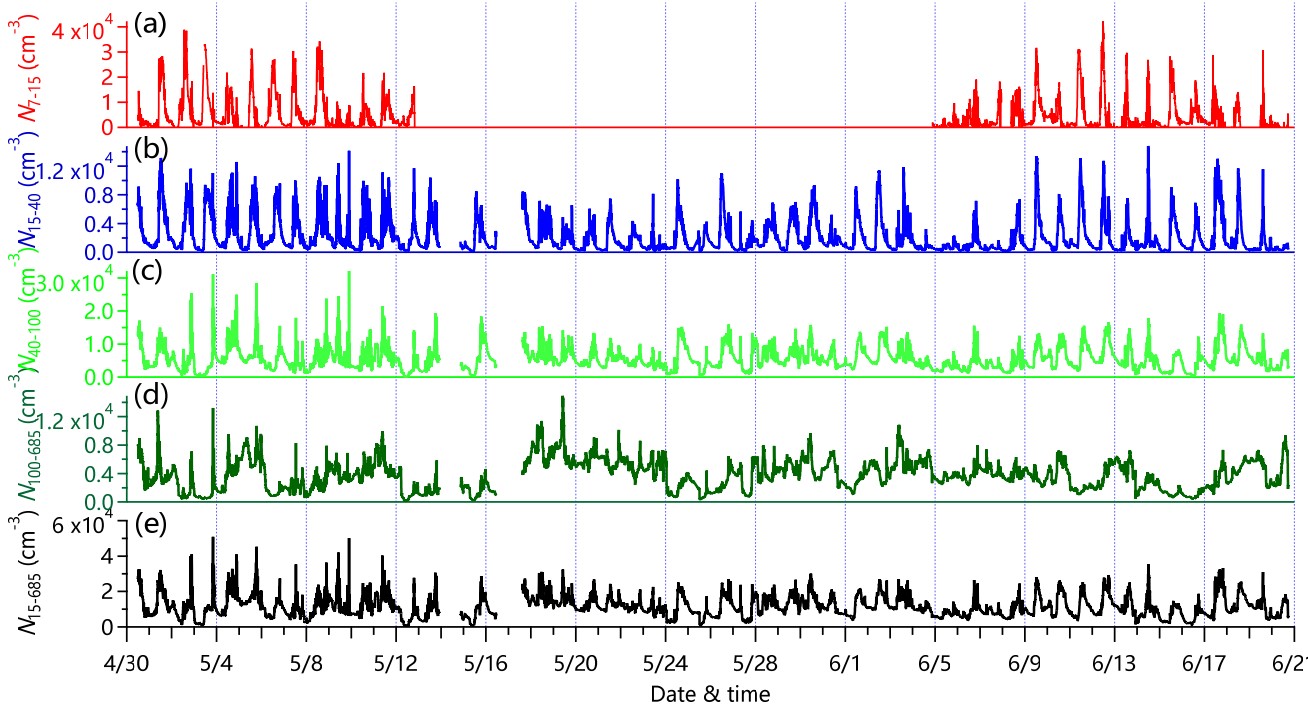

**Figure 6:** Time series of particle number concentrations for (a) $N_{7-15}$ calculated from the differences between MCPC and SMPS measurements, (b) $N_{15-40}$ (15–40 nm), (c) $N_{40-100}$ (40–100 nm), (d) $N_{100-685}$ (100–685 nm), and (e) all particles, $N_{15-685}$ (15–685 nm). The gap in (a) is mainly due to the malfunction of MCPC during this period.

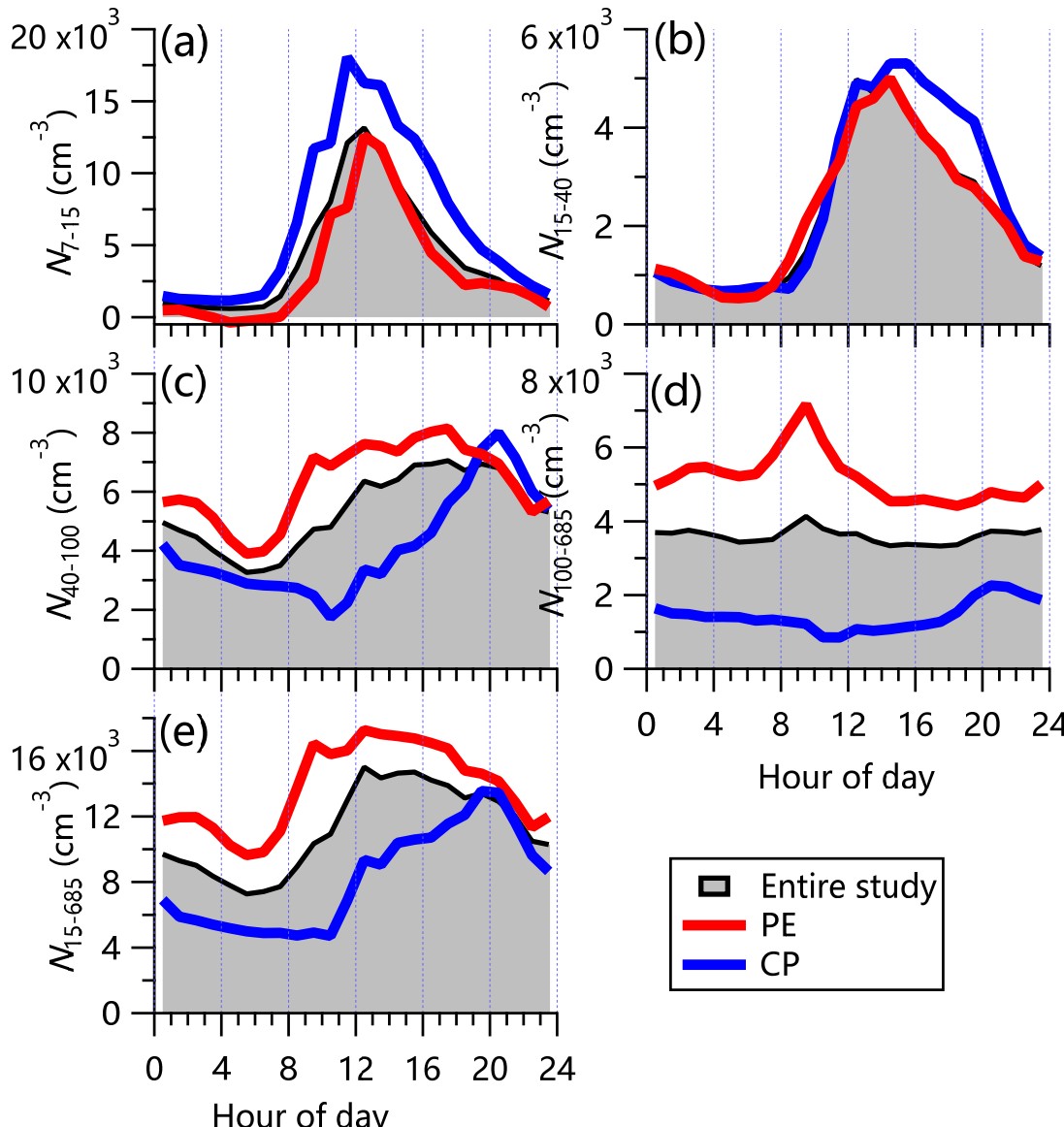

**Figure 7:** Diurnal cycles of particle number concentration for (a) $N_{7-15}$ calculated from the differences between MCPC and SMPS measurements, (b) $N_{15-40}$ (15–40 nm), (c) $N_{40-100}$ (40–100 nm), (d) $N_{100-685}$ (100–685 nm), and (e) all particles, $N_{15-685}$ (15–685 nm). Overall mean cycles are shown as black lines. Mean cycles for polluted events (PE) and clear periods (CP) are shown as red and blues lines, respectively.

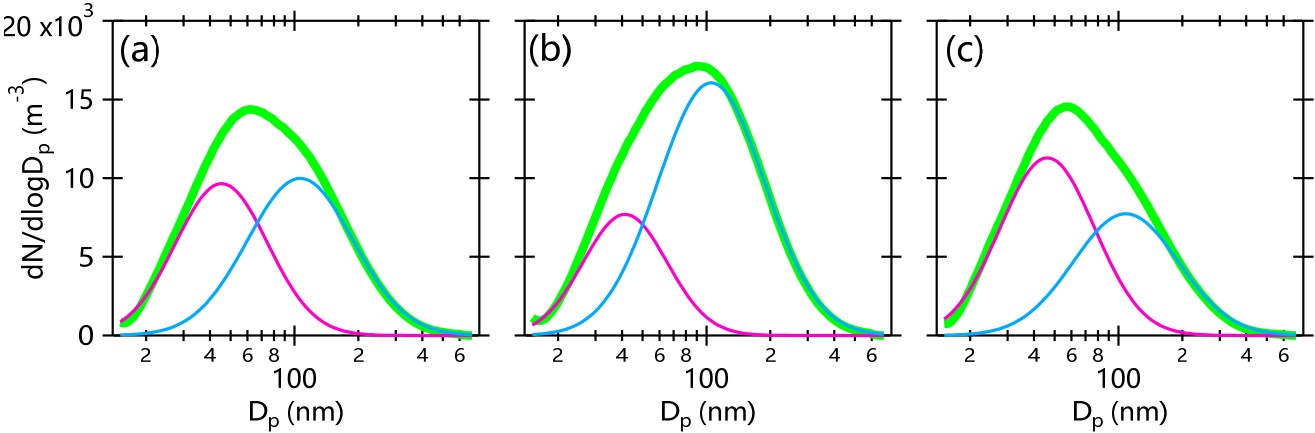

**Figure 8:** Average particle number size distributions during (a) the entire study, (b) polluted events, and (c) clean periods.

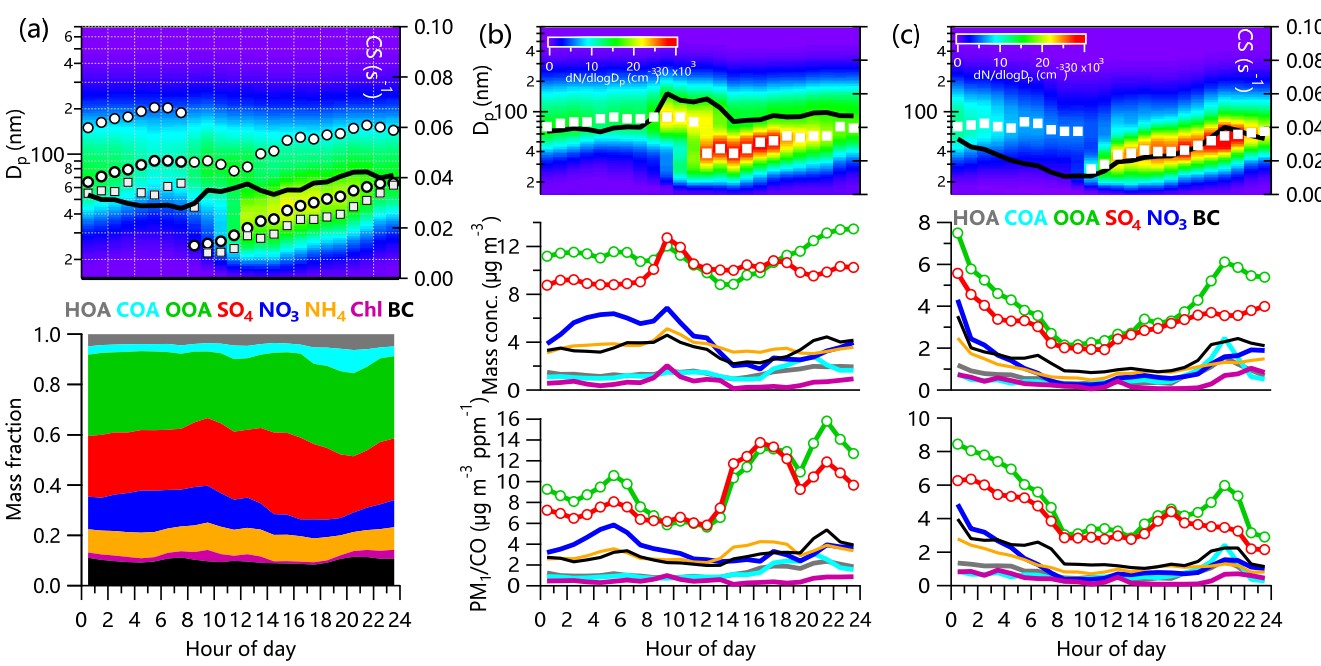

**Figure 9:** Average diurnal evolution of particle number size distributions and aerosol composition for new particle growth events during (a) the entire study, (b) polluted events, and (c) clean periods. The black solid lines in the top three panels show the diurnal cycles of CS. The circles and squares show the GMD from the log-normal fitting from this study and in Beijing (Du et al., 2017), respectively. The average diurnal cycles of aerosol species and CO-normalized aerosol species during
polluted events and clean periods are shown in (b) and (c).

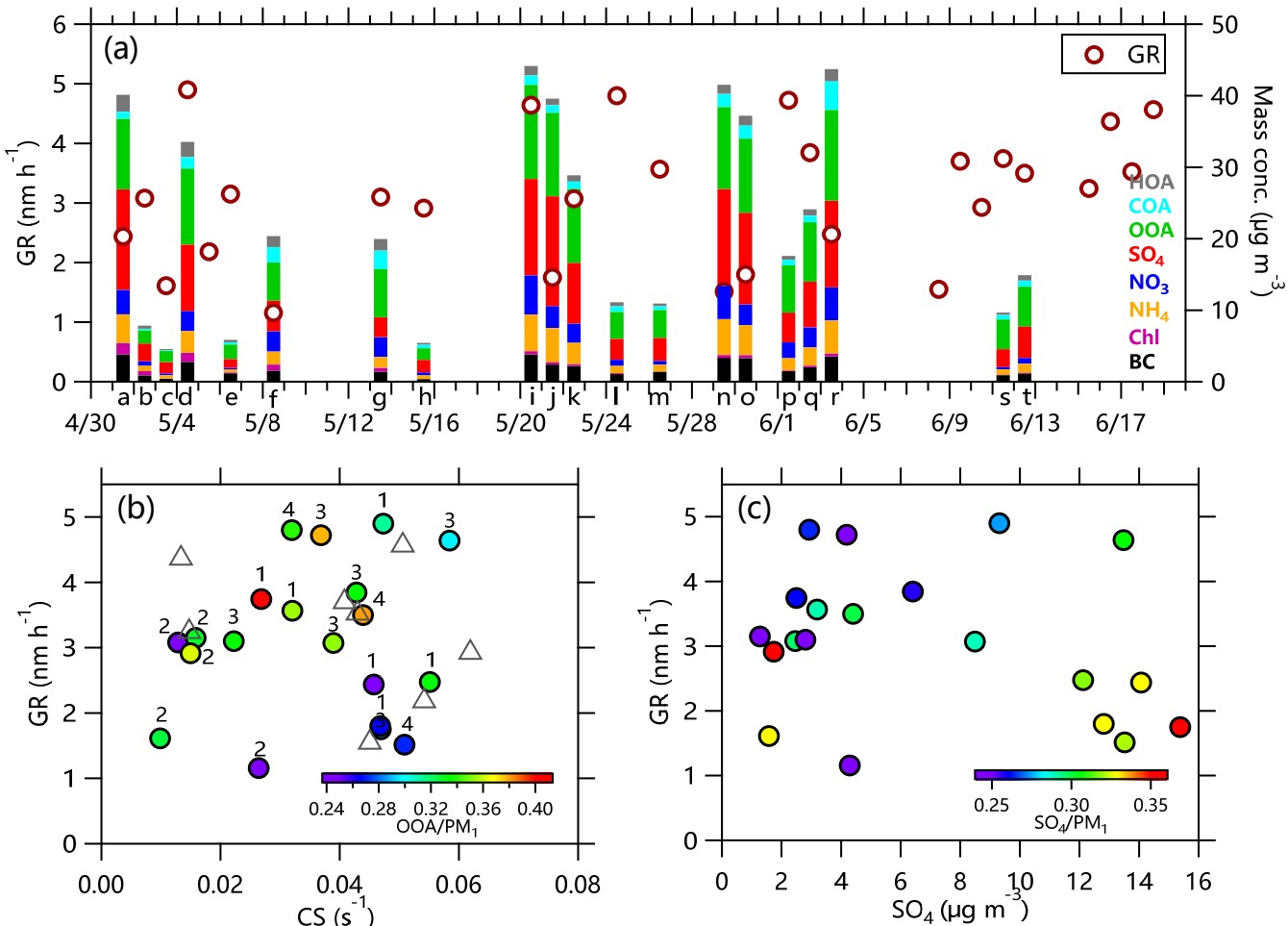

**Figure 10:** (a) Particle growth rates (GR) and the corresponding aerosol composition during the growth period, (b) the relationship between GR and CS, color-coded by $OOA/PM_1$, and (c) the relationship between GR and sulfate concentration, color-coded by $SO_4/PM_1$. The numbers over the circles in (b) represent different source regions (Fig. 11) and the triangles represent data without information about composition.

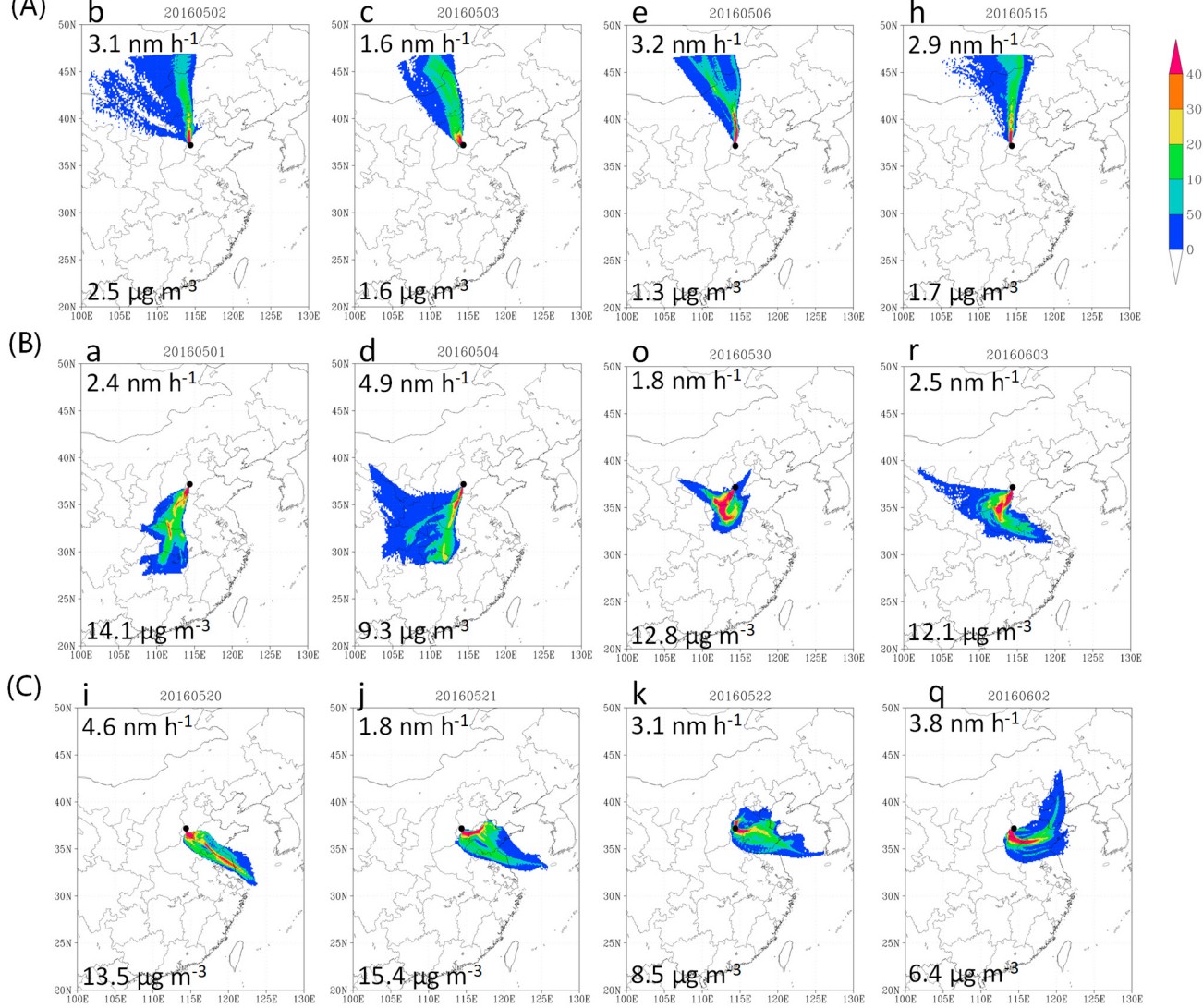

**Figure 11:** Particle GR and sulfate concentrations during the days shown in Fig. 10a with different footprints: (A) north, (B) southwest, and (C) east-southeast.

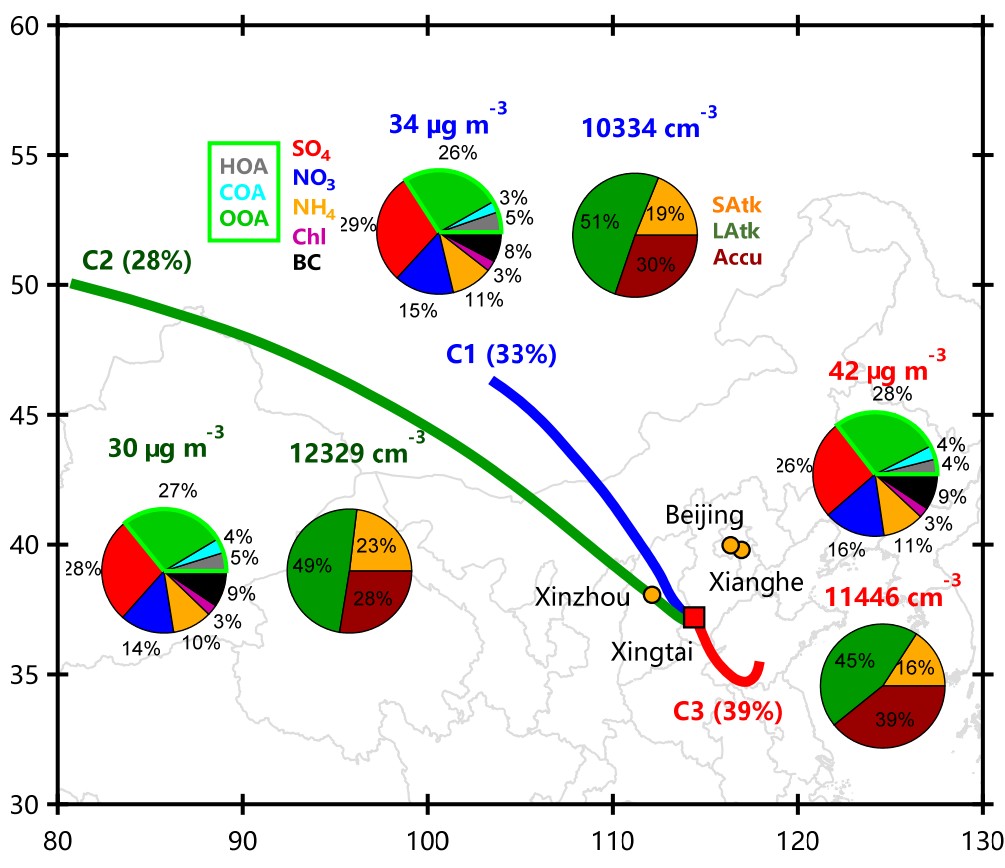

**Figure 12:** Average composition of aerosol particles and particle number concentration for three different clusters.