# Peer review of "Aerosol chemistry and particle growth events at an urban downwind site in North China Plain"

_Atmospheric Chemistry and Physics, 2017_

## Referee Comment (RC1) · Anonymous Referee #2 · 19 Feb 2018

The manuscript presented by Zhang et al. proposes an interesting study on the chemical composition and new particle formation in North China Plain. The characterization of PM1 sampled at the field site was performed using an aerosol mass spectrometer. Overall the work performed in this study is good and fall within the scope of the journal. However, the conclusions proposed from the PMF analysis are not always well sustained and more caution should be taken when extrapolating the results. Overall, more information needs to be added to validate PMF analysis and thus the conclusions of this study.

Lines 106:108: The molar ratio is not sufficient to predict aerosol acidity. Please use a thermodynamic model if you want to discuss aerosol acidity (e.g. Weber et al. 2016, Nature Geo.). Presence of organics also impacts CE. Overall the CE correction has to

be better constrained/explained. Affct í affect (line 108)

Lines 113-115: How does the aerosol density compare with other field measurements? How did the authors evaluate that 75% of the mass is within the mass range of 15-685 nm? What is the size cut off of the aethalometer? The authors should better explain how they considered the size ranges and how they compared particle mass between instruments: ACSM mass between 50-1000nm Aethalometer? SMPS 15-685 nm

In addition, did the authors correct their data by determining the transmission/losses of the particles into the different instruments?

Paragraph 2.3.2. This section should be moved into the "Results and discussion" section as the authors started to discuss the results. In addition, several questions remain in their data analysis and the authors should provide more information (i.e. robust validation).

Why did the authors directly constrain the PMF? Have they tried to constrain the PMF with biogenic factors (i.e. isoprene and/or monoterpene)? It is unclear why the authors were not able to obtain similar factors as previous studies (e.g. Fig S8). How do the factors correlate with the reference MS? How do the factors correlate throughout the campaign (i.e event vs non-event)? How do the residuals evolve throughout the campaign and especially during events vs non-events? It is a bit surprising to identify only 3 factors, assuming the complexity of the aerosol formation in such area. Especially when the authors argue that SOA formation is due to different air masses. Therefore, the authors should present the time series of the contribution of the factors and Q/Qexp parameter values across different model solutions as functions of the number of factors and constraint parameter (ðİŚŐ-value).

Lines 131:132: How is it an evidence that OOA is a surrogate of SOA? In addition, x vs y plot should be proposed for the entire period and not only for a selected period.

Lines 133:135: Show these results. It would help to understand/validate PMF analysis.

Lines 152:153: Add the concentrations of O3 & NOx to Figure 3

Line 155: It is clear for the period early June but it is not obvious for the other periods. RH stays similar and particle scavenging is not showed by the SMPS compared to the beginning of June. Please comment.

Lines 156:157 The peaks are very sudden and appear suspicious. How long do they last? In addition, those spikes do not show up in the mass fraction plot while they should. What is the reason? In some case, organics should explain 80-90% of the mass (e.g. early May) but the contribution of organics stays $\sim$ constant at 35-40%.

Lines 159:160 How do the diurnal cycles look like for those species? If it is only regional, why does the concentration of sulfate increase continuously? Shouldn't the authors expect to see a stronger diurnal variation if the wind changed?

Lines 160:161 It is hard to see as it is. The authors should plot the [NO3] vs the temperature.

Lines 162:163 The authors indicate that "These results suggest that urban downwind sites in the NCP experience similar PM pollution events as those in urban cities." How do the authors reach this conclusion?

Lines 184:187 & 189:190 If the authors claim that they observed two different air masses. The PMF should be able to distinguish clean vs haze events and differences should be visible in the MS.

Lines 240:242 How do the authors define a NPE?

Line 249: In general, the authors should present the x vs y plots when they want to correlate/demonstrate a correlation between two parameters.

Lines 260:265 What is the influence of biogenic-derived SOA during clean periods? Is it possible that biogenic-derived SOA contribute more to SOA formation yielding larger NPF? A better comparison of the MS and PMF data is needed to better understand

this aspect. In addition, what is the source of the big particles early morning during the polluted events? How do the meteorological conditions impact NPF?

---

## Referee Comment (RC2) · Anonymous Referee #1 · 24 Mar 2018

The authors reported the chemical composition and particle growth event at a downwind ground site in the North China Plain. The results mainly focus on analyzing the aerosol chemical composition measured by aerosol chemical speciation monitor (ACSM) and particle number concentration and size distribution obtained based on scanning mobility particle sizer (SMPS). OA sources (cooking OA, oxygenated OA, and hydrocarbon-like OA) were also resolved by combining source apportionment techniques with ACSM dataset. The species contribution (SO4 and OOA) are the main for particle growth has been analyzed, which is not new since most of the results have already been reported from multiple other field studies in NCP. The CE calculation and the way that the authors analyzed the correlation between chemical component in aerosols and growth rate have their own caveats, as addressed in the comments

following. Based on all the major comments as following, I recommend a major revision for this paper: Major comments: 1 The detailed operation of AMS and SMPS is totally missed in the paper. No information on how the ACSM were calibrated. 2 The CE needs to be re-evaluated. I have already raised this question in the quick review, however, the authors did not revise it. A CE of 0.51 and 0.57 was estimated based on equation in Middlebrook et al. (2012), when the average mass concertation of main components in polluted and clean days in Table 1 were used. The higher CE than default value of 0.5 is mainly due to acidic effect on aerosol bounces in AMS. Thus, the mass concentration of AMS might be overestimated in this study. 3 Line 111: The statement that the calculate SMPS mass is only 75% of the total PM1 cannot not be used a supporting evidence for CE. Multiple other studies show a good comparison of mass concentration between AMS and integrated SMPS, as examples shown below: Y. Zhang, L. Tang, P. L. Croteau, O. Favez, Y. Sun, M. R. Canagaratna, Z. Wang, F. Couvidat, A. Albinet, H. Zhang, J. Sciare, A. S. H. Prévôt, J. T. Jayne, and D. R. Worsnop: Field characterization of the PM2.5 Aerosol Chemical Speciation Monitor: insights into the composition, sources, and processes of fine particles in eastern China, Atmos. Chem. Phys., 17, 14501-14517, 10.5194/acp-17-14501-2017, 2017. X. F. Huang, L. Y. He, M. Hu, M. R. Canagaratna, Y. Sun, Q. Zhang, T. Zhu, L. Xue, L. W. Zeng, X. G. Liu, Y. H. Zhang, J. T. Jayne, N. L. Ng, and D. R. Worsnop: Highly time-resolved chemical characterization of atmospheric submicron particles during 2008 Beijing Olympic Games using an Aerodyne High-Resolution Aerosol Mass Spectrometer, Atmos Chem Phys, 10, 8933-8945, DOI 10.5194/acp-10-8933-2010, 2010. P. F. DeCarlo, E. J. Dunlea, J. R. Kimmel, A. C. Aiken, D. Sueper, J. Crounse, P. O. Wennberg, L. Emmons, Y. Shinozuka, A. Clarke, J. Zhou, J. Tomlinson, D. R. Collins, D. Knapp, A. J. Weinheimer, D. D. Montzka, T. Campos, and J. L. Jimenez: Fast airborne aerosol size and chemistry measurements above Mexico City and Central Mexico during the MILAGRO campaign, Atmos Chem Phys, 8, 4027-4048, 2008. The authors should look more carefully into their dataset to find out the possible reasons that could cause this difference. 4 Line 200-215: The cooking OA (COA) and hydrocarbon-like OA (HOA) have

very similar diurnal variations, which show similar peaks in the noon and night time. It indicates that the PMF did not really separate the two factors. A detailed explanation on separating two factors are needed. 5 Line 291-292: please give statistic analysis result on correlation of Fig. 10 b-c to prove the conclusion stated in this study. I did not see the correlation between growth rate and condensation sink. Minor comments: Line 56-57: Give references. Why did the mass concentration of PM1 in Xingtai ($\sim$30 $\mu$g/m3) is so low compared with the measured annul mass concentration ($\sim$90 $\mu$g/m3) in the past. Line 108: Have the authors done the RIE calibration of NH4 and SO4. If so, please specify. Line 129: Please give the range of f55 vs f57 in COA in Mohr et al. 2012. Line 271: Give detailed evidence for this sentence. Line 277: Please specify if the CO background is offset here. If not, the background of CO should be deducted in the calculation. For an example, J. A. de Gouw, D. Welsh-Bon, C. Warneke, W. C. Kuster, L. Alexander, A. K. Baker, A. J. Beyersdorf, D. R. Blake, M. Canagaratna, A. T. Celada, L. G. Huey, W. Junkermann, T. B. Onasch, A. Salcido, S. J. Sjostedt, A. P. Sullivan, D. J. Tanner, O. Vargas, R. J. Weber, D. R. Worsnop, X. Y. Yu, and R. Zaveri: Emission and chemistry of organic carbon in the gas and aerosol phase at a sub-urban site near Mexico City in March 2006 during the MILAGRO study, Atmos Chem Phys, 9, 3425-3442, 2009. Line 290: No information on how the growth rate was calculated. Line 291: What is the GR range in the literatures. How does is compare to the results form Beijing?

References: Ann M. Middlebrook, Roya Bahreini, Jose L. Jimenez, and Manjula R. Canagaratna: Evaluation of Composition-Dependent Collection Efficiencies for the Aerodyne Aerosol Mass Spectrometer using Field Data, Aerosol Sci Tech, 46, 258-271, 10.1080/02786826.2011.620041, 2012.

---

## Author Comment (AC1) · 1 Jun 2018

We are thankful to the two reviewers for their thoughtful comments that help improve the manuscript significantly. Following the reviewers' suggestions, we have revised the manuscript accordingly. Listed below are our point-by-point responses in blue to each reviewer's comments.

**Response to reviewer #1**

The authors reported the chemical composition and particle growth event at a downwind ground site in the North China Plain. The results mainly focus on analyzing the aerosol chemical composition measured by aerosol chemical speciation monitor (ACSM) and particle number concentration and size distribution obtained based on scanning mobility particle sizer (SMPS). OA sources (cooking OA, oxygenated OA, and hydrocarbon-like OA) were also resolved by combining source apportionment techniques with ACSM dataset. The species contribution (SO4 and OOA) are the main for particle growth has been analyzed, which is not new since most of the results have already been reported from multiple other field studies in NCP. The CE calculation and the way that the authors analyzed the correlation between chemical component in aerosols and growth rate have their own caveats, as addressed in the comments following. Based on all the major comments as following, I recommend a major revision for this paper.

We thank the reviewer's comments.

Major comments:

1. The detailed operation of AMS and SMPS is totally missed in the paper. No information on how the ACSM were calibrated.

We thank the reviewer's comments. The detailed operations of ACSM and SMPS were now added in the revised manuscript.

"A PM$_{2.5}$ cyclone (Model: URG-2000-30ED) was supplied in front of the sampling inlet to remove coarse particles larger than 2.5 μm. The ambient air was drawn into the container through a 1/2 inch (outer diameter) stainless steel tube at a flow rate of 3 L min$^{-1}$ using an external pump, of which ~0.1 L min$^{-1}$ was sub-sampled into the ACSM. The sampling height was approximately 2 m, which was 1.5 m higher than the roof of container. Thus, the particle residence time in the sampling tube was about 5 s. Aerosol particles were then dried by a silica gel diffusion dryer before sampling into the ACSM. Before the campaign, the ACSM was calibrated with pure ammonium nitrate particles following the standard protocols in Ng et al. (2011b).

"The size-resolved particle number concentration in the size range from 15 to 685 nm was measured in situ by a condensation particle counter (CPC, model 3775, TSI) equipped with a long differential mobility analyser (DMA, model 3081A, TSI). The time resolution is 5 min."

2. The CE needs to be reevaluated. I have already raised this question in the quick review, however, the authors did not revise it. A CE of 0.51 and 0.57 was estimated based on equation in Middlebrook et al. (2012), when the average mass concentration of main components in polluted and clean days in Table 1 were used. The higher CE than default value of 0.5 is mainly due to acidic effect on aerosol bounces in AMS. Thus, the mass concentration of AMS might be overestimated in this study.

Thank the reviewer's comments. Following the reviewer's comments, we recalculated mass concentration of NR-PM$_1$ species using composition-dependent CE (average: 0.5036) as follows (Middlebrook et al., 2012):

$$CE_{dry} = \max\left[0.45, 1.0 - 0.73 \times \left(NH_4/NH_{4,predict}\right)\right]$$

The average mass concentration of Org, $SO_4$, $NO_3$, $NH_4$, Chl were 11.5, 7.4, 4.2, 3.0, and 0.8 µg m$^{-3}$ respectively, which were almost the same as the results calculated by CE = 0.5 (Table R1).

Table R1. The average mass concentration of Org, $SO_4$, $NO_3$, $NH_4$, Chl measured by ACSM using different CE values.

|  | CE = 0.5 | CE algorithm |
|---|---|---|
| Org | 11.8 ± 7.3 | 11.5 ± 7.3 |
| $SO_4$ | 7.7 ± 5.1 | 7.4 ± 5.2 |
| $NO_3$ | 4.3 ± 4.3 | 4.2 ± 4.3 |
| $NH_4$ | 3.1 ± 2.3 | 3.0 ± 2.4 |
| Chl | 0.8 ± 1.4 | 0.8 ± 1.2 |

We had carefully considered the reviewer's comments. CE was introduced to correct the incomplete detection of submicron particles by ACSM. It depended on particle acidity, ammonium nitrate fraction and relative humidity. In this study, 1) aerosol particles were not acidic enough to affect the CE substantially. 2) The average mass fraction of $NH_4NO_3$ was 18%, which would not affect CE substantially. 3) A silica gel diffusion dryer was used to dry aerosol particles before sampling into ACSM.

Thus, we kept CE = 0.5, and the results were comparable within the acceptable difference.

3. Line 111: The statement that the calculate SMPS mass is only 75% of the total $PM_1$ cannot not be used a supporting evidence for CE. Multiple other studies show a good

comparison of mass concentration between AMS and integrated SMPS, as examples shown below: Y. Zhang, L. Tang, P. L. Croteau, O. Favez, Y. Sun, M. R. Canagaratna, Z. Wang, F. Couvidat, A. Albinet, H. Zhang, J. Sciare, A. S. H. Prévôt, J. T. Jayne, and D. R. Worsnop: Field characterization of the PM2.5 Aerosol Chemical Speciation Monitor: insights into the composition, sources, and processes of fine particles in eastern China, Atmos. Chem. Phys., 17, 14501-14517, 10.5194/acp-17-14501-2017, 2017. X. F. Huang, L. Y. He, M. Hu, M. R. Canagaratna, Y. Sun, Q. Zhang, T. Zhu, L. Xue, L. W. Zeng, X. G. Liu, Y. H. Zhang, J. T. Jayne, N. L. Ng, and D. R. Worsnop: Highly time–resolved chemical characterization of atmospheric submicron particles during 2008 Beijing Olympic Games using an Aerodyne High-Resolution Aerosol Mass Spectrometer, Atmos Chem Phys, 10, 8933-8945, DOI 10.5194/acp-10-8933-2010, 2010. P. F. DeCarlo, E. J. Dunlea, J. R. Kimmel, A. C. Aiken, D. Sueper, J. Crounse, P. O. Wennberg, L. Emmons, Y. Shinozuka, A. Clarke, J. Zhou, J. Tomlinson, D. R. Collins, D. Knapp, A. J. Weinheimer, D. D. Montzka, T. Campos, and J. L. Jimenez: Fast airborne aerosol size and chemistry measurements above Mexico City and Central Mexico during the MILAGRO campaign, Atmos Chem Phys, 8, 4027-4048, 2008. The authors should look more carefully into their dataset to find out the possible reasons that could cause this difference.

We agree with the reviewer's comments, and modified the inappropriate explanation in the revised manuscript. In fact, such a difference can be caused by (1) measurement uncertainties between different instruments. For example, the SMPS measurement uncertainties can be increased to 30% for particles larger than 200 nm, which dominated the total particle mass (Wiedensohler et al., 2012), while the uncertainties for ACSM measurements varied from 9 – 36% for different aerosol species (Crenn et al., 2015), (2) the effects of particle shape. In this study, we assume spherical particles, and aerodynamic diameter ($D_{va}$) is approximately equal to mobility diameter ($D_m$) times particle density, and (3) the uncertainties in estimating particle density because we didn't the measurements of refractory species, e.g., mineral elements. Similar difference in the comparisons between AMS and SMPS

were also observed in many previous studies in China (Zhang et al., 2011; Xu et al., 2014).

4. Line 200-215: The cooking OA (COA) and hydrocarbon-like OA (HOA) have very similar diurnal variations, which show similar peaks in the noon and night time. It indicates that the PMF did not really separate the two factors. A detailed explanation on separating two factors are needed.

Thanks for the reviewer's comments. It is difficult to separate HOA from COA by PMF using quadrupole AMS or ACSM measurements (Sun et al., 2010; Sun et al., 2012). Thus, source apportionment using the bilinear model through a multilinear engine (ME-2) was applied to non-refractory organic aerosol mass spectra collected by ACSM. ME-2 is capable of finding acceptable solutions in accordance with the constraints provided by the user. In this study, we input HOA and COA reference profiles with a-values varying from 0 to 1. The a-value determines the extent to which the output factor profiles is allowed to vary from the input factor profiles. An optimal solution involving three factors with an a-value of 0.2 was accepted at last.

In order to examine the rationality of the solutions, we then tried to use BC as a tracer to separate HOA from COA assuming that BC is dominantly from traffic emissions while the contribution from cooking emissions is minor (Pei et al., 2016). In this study, POA was highly correlated ($r^2 > 0.66$) with BC between 1:00 – 10:00 when cooking emissions are not significant (Figure R1). The ratios of POA/BC were also the lowest during this period, suggesting the dominant contribution of HOA to POA. The average ratio of POA/BC during the period without significant cooking emissions is 0.62. Thus, we estimated the mass concentrations of HOA and COA following the equations below:

(1)  $HOA = (POA/BC)_{nc} \times BC - COA_b$

(2)  $COA = POA - HOA$

(POA/BC) is the average ratio of POA/BC during the periods without significant cooking emissions, which is 0.62 here, and $COA_b$ is the background concentration of cooking aerosols. According to the diurnal cycles of COA in previous studies in Beijing (Huang et al., 2010; Hu et al., 2016; Hu et al., 2017), we found a background concentration of 1.30 ± 0.39 μg m$^{-3}$. In view of the fact that there are fewer local cooking emissions compared with Beijing, we hypothesized a background concentration of 0.7 μg m$^{-3}$ (half of that in Beijing) for the estimation in this study.

[Figure]

Figure R1. Diurnal variations of POA, BC, correlation coefficients and slopes of POA vs. BC.

[Figure]

Figure R2. Scatter plot of POA vs. BC, and the red dots are the data between 1:00 – 10:00.

The BC-tracer method has been used in Sun et al. (2018) to response the mix of HOA and COA. The diurnal cycles of estimated COA and is also consistent with the results of ME-2 analysis (Figure R3). The estimated HOA and COA on average contributed both 11% to OA, consistent with the results of ME-2 analysis. This suggested that the results from ME-2 analysis are reasonable.

[Figure]

Figure R3. Diurnal cycles of estimated COA and HOA using BC as a tracer.

5. Line 291-292: please give statistic analysis result on correlation of Fig. 10b-c to prove the conclusion stated in this study. I did not see the correlation between growth rate and condensation sink.

Thanks for your comments. We revised this paragraph following the reviewer's comments. The statistic analysis results (value of $r^2$) and plots are also added in Section 3.4 and supplementary materials.

"As indicated in Fig. 10b, GR was positively correlated with CS for most of the time, in particular when the fraction of OOA is higher than 30% (Fig. R4, $r^2 = 0.61$). As CS increased from ~ 0.01 $s^{-1}$ to 0.05 $s^{-1}$, GR increased from 3 nm $h^{-1}$ to 5 nm $h^{-1}$. However, there were some low GRs with high CS, which were characterized by low contributions of OOA. These results show the importance of the involvement of OOA in particle growth. Figure 10c further shows that GR was positively correlated with the concentration of sulfate during periods with low sulfate mass loadings (Fig. R5, < 3 μg $m^{-3}$, $r^2=0.42$), while periods with higher concentrations of sulfate had lower GRs."

[Figure]

Figure R4. The relationship between GR and CS in the OOA fraction of more than 32% conditions.

[Figure]

Figure R5. The relationship between GR and sulfate concentration in the sulfate concentration of less than 3 µg m$^{-3}$ conditions.

Minor comments:

1. Line 56-57: Give references. Why did the mass concentration of PM$_1$ in Xingtai (~30 µg m$^{-3}$) is so low compared with the measured annual mass concentration (~90 µg m$^{-3}$) in the past.

Thanks for your comments. The source of data was added in the revised manuscript.

"(the data are from four monitoring sites in urban Xingtai that was released by the China National Environmental Monitoring Centre)"

The mass concentration of PM$_1$ in this study (~30 µg m$^{-3}$) is so low compared with the measured annual mass concentration (~90 µg m$^{-3}$) in the past. This is mainly due to three reasons: (1) In this study, the mass concentration of PM$_1$ was collected by ACSM and AE33 from 30 April to 20 June. The pollution during this period was relatively light compared to that in winter. (2) The results reported in this study are

PM$_1$ and those reported in the past are PM$_{2.5}$. (3) The sample site in this study is a suburban site. The pollution level is lower than those at urban sites where high PM$_{2.5}$ values were observed.

2. Line 108: Have the authors done the RIE calibration of NH4 and SO4. IF so, please specify.

Thanks for your comments. We did the RIE calibration of SO$_4$ using ammonium sulfate, and the RIE of SO$_4$ was 0.98. Such information was now added in the revised manuscript.

"Default relative ionization efficiencies (RIE) were used except ammonium (5.0) and sulphate (0.98) that were determined from pure ammonium nitrate and ammonium sulphate, respectively."

3. Line 129: Please give the range of f55 vs f57 in COA in Mohr et al. 2012.

Thanks for your comments. We added the range of f55 vs f57 in COA in Mohr et al. 2012.

"(~> 1.2)"

4. Line 271: Give detailed evidence for this sentence.

Thanks for your comments. Because the growth sizes of particles in this study (from ~ 25 nm to ~60 nm) are larger than those observed in urban Beijing (from ~22 nm to ~55 nm, Fig.9a), we speculated a stronger aging process (characterized by particle growth) at the suburban site. We revised the sentence as follows:

"Note that the growth sizes of particles were overall larger than those observed in urban Beijing (from ~22 nm to ~55 nm), likely indicating a stronger aging process at the suburban site."

5. Line 277: Please specify if the CO background is offset here. If not, the background of CO should be deducted in the calculation. For an example, J. A. de Gouw, D. Welsh-Bon, C. Warneke, W. C. Kuster, L. Alexander, A. K. Baker, A. J. Beyersdorf, D. R. Blake, M. Canagaratna, A. T. Celada, L. G. Huey, W. Junkermann, T. B. Onasch, A. Salcido, S. J. Sjostedt, A. P. Sullivan, D. J. Tanner, O. Vargas, R. J. Weber, D. R. Worsnop, X. Y. Yu, and R. Zaveri: Emission and chemistry of organic carbon in the gas and aerosol phase at a sub-urban site near Mexico City in March 2006 during the MILAGRO study, Atmos Chem Phys, 9, 3425-3442, 2009.

Thanks for your comments. The CO background was subtracted. We clarified this in the revised manuscript. It now reads: "Note that CO here was subtracted by a background value of 0.068 ppm that was calculated as the average of the lowest 5% data in this study. "

6. Line 290: No information on how the growth rate was calculated.

Thanks for your comments. We added it in the revised manuscript.

"The particle growth rates were calculated using Eq. (1).

$$GR = \frac{\Delta D_m}{\Delta t} \tag{1}$$

Where $D_m$ is the geometric mean diameter from the log-normal fitting of each size distribution and $\Delta D_m$ is the increase in diameter during the growth period of $\Delta t$."

7. Line 291: What is the GR range in the literatures. How does is compare to the results from Beijing?

Thanks for your comments. We added the GR range from the literatures in the revised manuscript.

"(1-20 nm h$^{-1}$)"

Compared to the results in Beijing (approximate 1.6-6 nm h$^{-1}$), particle growth rates in this study appear to be faster.

**Response to reviewer #2**

The manuscript presented by Zhang et al. proposes an interesting study on the chemical composition and new particle formation in North China Plain. The characterization of PM1 sampled at the field site was performed using an aerosol mass spectrometer. Overall the work performed in this study is good and fall within the scope of the journal. However, the conclusions proposed from the PMF analysis are not always well sustained and more caution should be taken when extrapolating the results. Overall, more information needs to be added to validate PMF analysis and thus the conclusions of this study.

We thank the reviewer's positive comments.

Comments:

1. Line 106-108: The molar ratio is not sufficient to predict aerosol acidity. Please use a thermodynamic model if you want to discuss aerosol acidity (e.g., Weber et al. 2016, Nature Geo.). Presence of organics also impacts CE. Overall the CE correction has to be better constrained/explained. Affct should be affect (line 108).

We agree with your comments. Unfortunately, the measurements of gaseous $HNO_3$, HCl and $NH_3$ were not available in this study. So a thermodynamic model may be not suitable to predict aerosol acidity. And molar ratio can reflect aerosol acidity to some extent, despite the large uncertainties. We just use molar ratio to explain the limited influence on CE from aerosol acidity qualitatively.

Most importantly, Middlebrook et al. (2012) has a full evaluation of AMS CE and found that CE is mainly affected by three factors, i.e., particle acidity (measured $NH_4^+$ vs. predicted $NH_4^+$), RH, and fraction of ammonium nitrate. These three factors were all addressed in our study, and a final of CE of 0.5 is suitable for the entire campaign. Also see our response to reviewer #1.

"Affct " was corrected as "affect".

2. Line 113-115: How does the aerosol density compare with other field measurements? How did the authors evaluate that 75% of the mass is within the mass range of 15-685 nm? What is the size cut off of the aethalometer? The authors should better explain how they considered the size ranges and how they compared particle mass between instruments: ACSM mass between 50-1000 nm Aethalometer? SMPS 15-685 nm

We thank the reviewer's comments. The average aerosol density in this study was 1.5 g cm$^{-3}$, consistent with that in Beijing (1.2-1.6 g cm$^{-3}$, Zhao et al., 2017), Changdao (1.5 g cm$^{-3}$, Hu et al., 2013), Lanzhou (1.61 g cm$^{-3}$, Xu et al., 2014).

A PM$_{2.5}$ cyclone was supplied in front of the Aethalometer sampling inlet to remove coarse particles larger than 2.5 µm. The ratio of the vacuum aerodynamic diameter ($D_{va}$) measured by ACSM/mobility diameter ($D_p$) measured by SMPS is a function of particle shape and density. Assuming spherical particles, $D_{va}$ is approximately equal to $D_p$ × density. Therefore, $D_p$ measured by SMPS is in the range of 22-1027 nm in $D_{va}$, which is similar to that of ACSM. Because the ACSM does not measure BC, Aethalometer was used to measure BC. Although the size cutoff is 2.5 µm, BC from combustion emissions is dominantly in small particles, and the peak diameters are typically 100 – 200 nm.

In fact, such a difference between ACSM and SMPS can be caused by (1) measurement uncertainties between different instruments. For example, the SMPS measurement uncertainties can be increased to 30% for particles larger than 200 nm, which dominated the total particle mass (Wiedensohler et al., 2012), while the uncertainties for ACSM measurements varied from 9 – 36% for different aerosol species (Crenn et al., 2015), (2) the effects of particle shape. In this study, we assume spherical particles, and aerodynamic diameter ($D_{va}$) is approximately equal to mobility diameter ($D_m$) times particle density, and (3) the uncertainties in estimating

particle density because we didn't the measurements of refractory species, e.g., mineral elements. Similar differences in the comparisons between AMS and SMPS were also observed in many previous studies in China (Zhang et al., 2011; Xu et al., 2014).

In addition, did the authors correct their data by determining the transmission/losses of the particles into the different instruments?

Thanks for your comments. Ambient air was drawn into the container through a 1/2 inch (outer diameter) stainless steel tube (about 2 m) at a flow rate of 3 L min$^{-1}$. The residence time of aerosol in the sampling tube is approximately 5 seconds. The losses for particles between 20 – 1000 nm are less than 5%.

3. Paragraph 2.3.2. This section should be moved into the "Results and discussion" section as the authors started to discuss the results. In addition, several questions remain in their data analysis and the authors should provide more information (i.e. robust validation).

Thanks for your comments. The content in paragraph 2.3.2 was mainly focused on source apportionment method and its rationality. So we kept it in this section and added more information to validate our results in the response.

"We also noticed that HOA showed similar diurnal cycle as that of COA even though they were separated in ME-2 analysis. To better clarify the uncertainties, we estimated COA concentrations using BC as a tracer for HOA. Indeed, POA was highly correlated ($r^2$ > 0.66) with BC between 1:00 and 10:00 when cooking emissions are little, suggesting the dominant contribution of traffic emissions on BC. The average ratio of POA/BC during this period (0.62) was then used to estimate HOA, and COA was estimated based on the difference between POA and HOA. Our results indicated that COA and HOA concentrations estimated using BC-tracer method on average both contributed 11% to OA which is well consistent with those using ME-2 analysis.

These results together imply that the results of ME-2 analysis are relatively reasonable."

4. Why did the authors directly constrain the PMF? Have they tried to constrain the PMF with biogenic factors (i.e. isoprene and/or monoterpene)? It is unclear why the authors were not able to obtain similar factors as previous studies (e.g. Fig S8). How do the factors correlate with the reference MS? How do the factors correlate throughout the campaign (i.e. event vs non-event)? How do the residuals evolve throughout the campaign and especially during events vs non-events? It is a bit surprising to identify only 3 factors, assuming the complexity of the aerosol formation in such area. Especially when the authors argue that SOA formation is due to different air masses. Therefore the authors should present the time series of the contribution of the factors and Q/Qexp parameter values across different model solutions as functions of the number of factors and constraint parameter (ISO-value).

Thanks for your comments. Quadrupole ACSM is still limited compared with the research-grade AMS, in terms of sensitivity and mass resolution. Also, ACSM introduces more uncertainties because of ion transmission efficiency between m/z 50 – 150. Therefore, PMF analysis of ACSM is very difficult to identify more than three factors. Previous researches using quadrupole ACSM in the NCP only resolved two factors, i.e., POA and SOA (Chen et al., 2015; Sun et al., 2012; Jiang et al., 2013). Because there are restaurants nearby, we are sure that our sampling site has somewhat cooking influences. Thus, we uses ME-2 analysis to separate HOA and COA. The BC-tracer method further showed that the COA results is reasonable, as shown in our response to the fourth comment of reviewer #1. The relationship between factors MS and the reference MS are shown in Fig. R6 and R7. The time series of the contribution of the factors are shown in Fig. R8.

[Figure]

Figure R6. The relationship between COA MS and the reference MS.

[Figure]

Figure R7. The relationship between HOA MS and the reference MS.

[Figure]

Figure R8. The time series of the contribution of the factors.

5. Lines 131-132: How is it an evidence that OOA is a surrogate of SOA? In addition, x vs y plot should be proposed for the entire period and not only for a selected period.

Thanks for your comments. The mass spectrum of OOA is characterized by a prominent peak at *m/z* 44. And OOA was highly correlated with SIA, suggesting its secondary source to some extent. Previous studies have shown that OOA was highly correlated with secondary organic carbon everywhere, and OOA is a good surrogate of SOA (Zhang et al., 2007).

Figure 2 has been proposed for the entire period and the missed data between 14-20, June is due to the malfunction of the instruments.

6. Lines 133-135: Show these results. It would help to understand/validate PMF analysis.

Thanks for your comments. Results of two-factor and three-factor resolved by PMF

are shown in Figure R9 and R10. PMF can identify two-factor solution using data from quadrupole ACSM, the solutions with three and more factors show a splitting and mixing of factors.

[Figure]

FigureR9. Mass spectral profiles (on the right) and time series of the mass concentration of two OA factors (on the left).

[Figure]

FigureR10. Mass spectral profiles (on the right) and time series of the mass concentration of three OA factors (on the left).

7. Lines 152-153: Add the concentrations of O3 & NOx to Figure 3.

Thanks for your comments. In the beginning, we prepared to add the time series of gaseous pollutants to Figure 3. But another team in our project who is responsible for gaseous measurement wanted to use them to write another paper. To avoid the conflict, we didn't show time series of gaseous species in Figure 3. We plot Figure S11 to help analyze gaseous pollutants and its impact on PM.

8. Line 155: It is clear for the period early June but it is not obvious for the other periods. RH stays similar and particle scavenging is not showed by the SMPS compared to the beginning of June. Please comment.

Thanks for your comments. We determined wet scavenging through precipitation data, although the RH was not that higher. Particle scavenging is not obvious from the SMPS measurements because of the high concentration of Aitken mode particles. In fact, the number concentration of accumulation mode particles that contributed to particle mass decreased significantly corresponding to particle scavenging (Fig. 6d). To avoid misreading, we revised the sentence as "Several pollution episodes usually lasting ~2–3 days were observed during the study period, e.g., on 9-11, 17–23, 28–31 May, and 2–4 June. These pollution episodes were quickly cleaned mainly by wet scavenging."

9. Line 156-157: The peaks are very sudden and appear suspicious. How long do they last? In addition, those spikes do not show up in the mass fraction plot while they should. What is the reason? In some case, organics should explain 80-90% of the mass (e.g. early May) but the contribution of organics stays ~ constant at 35-40%.

Thanks for your comments. We checked the peaks and found they last only 5-10 min (one or two data points). They may be from instantly local emissions. In these cases, although organics had the highest mass concentration, other species also increased obviously. Thus, the contribution of organics was still lower than 60%.

10. Lines 159-160: How do the diurnal cycles look like for those species? If it is only regional, why does the concentration of sulfate increase continuously? Shouldn't the authors expect to see a stronger diurnal variation if the wind changed?

Good point. Sulfate concentrations obviously increased and remained relatively high concentration during the pollution events. In fact, we also found that the mass concentration of sulfate varied with the change of wind direction. In most cases, it increased when the wind from south while decreased when the wind from north-west. This result also showed in bivariate polar plots of sulfate as a function of wind speed and wind direction (Figure 5c). Thus, we concluded that sulfate was mainly influenced by regional transport although there was other formation mechanism. To avoid misreading, we revised the sentence as "By contrast, sulfate concentrations obviously increased and remained relatively high during the pollution events, suggesting the important role of regional transport at the downwind site of Xingtai."

11. Lines 160-161: It is hard to see as it is. The authors should plot [NO3] vs the temperature.

Thanks for your comments. The diurnal evolution of nitrate showed high concentration at night but low concentration during daytime, which was mainly driven by the gas particle partitioning due to the influence of temperature (Fig. 4d). Because the nitrate formation is also affected by daytime photochemical production (OH + NO$_2$) and nighttime heterogeneous reactions, and also the temperature and precursors varied largely from day to day, you would not expect clear relationship between nitrate and temperature for all data points. The diurnal profile is one of the best ways to show the temperature-influenced gas-particle partitioning.

12. Lines 162-163: The authors indicate that "These results suggest that urban downwind sites in the NCP experience similar PM pollution events as those in urban cities." How do the authors reach this conclusion?

Thanks for your comments. We thought the results of sampling site exceeded the Chinese National Ambient Air Quality Standards by 29%. They also experienced atmospheric pollution. So we reached this conclusion. And we revised improper expression in the revised manuscript.

"The average $PM_{2.5}$ mass concentration was 45.2 μg m$^{-3}$. Although the average $PM_{2.5}$ mass concentration was 15% lower than that (53.3 μg m$^{-3}$) measured at the urban sites in Xingtai, it exceeded the Chinese National Ambient Air Quality Standards by 29%. These results suggest that the urban downwind sites also experience similar PM pollution events as the urban sites."

13. Lines 184-187 & 189-190: If the authors claim that they observed two different air masses. The PMF should be able to distinguish clean vs haze events and differences should be visible in the MS.

Thanks for your comments. We agreed with the reviewer's comment. In Table 1, the mass concentrations of HOA, COA, and OOA were 1.66, 1.67, and 12.2 μg m$^{-3}$ during pollution periods, which were much higher than that during clean periods. The use of ME-2 forced the MS to some extent and reduced the differences between clean and haze events in the MS.

14. Lines 240-242: How do the authors define a NPE?

Thanks for your comments. New particle growth events (NPE) occurred in daytime were identified according to four comments: a) outbreak of number concentration of small Aitken mode particles ($N_{20-40}$), b) the number concentration of large Aitken mode and accumulation particles didn't increased in a consistent manner, c) the particle number size distribution showed a small size mode peaked at ~20 nm in the initial process, d) the geometric mean diameter (GMD) of new particle mode continued to grow more than 3 hours.

15. Line 249: In general, the authors should present the x vs y plots when they want to correlate/demonstrate a correlation between two parameters.

Thanks for your comments. As the reviewer suggested, x vs y plots have been used in the data analysis (Figure R11 and R12).

[Figure]

Figure R11. The relationship between $N_{40-100}$ and $N_{15-685}$.

[Figure]

Figure R12. The relationship between $V_{40-100}$ and $V_{15-685}$.

16. Lines 260-265: What is the influence of biogenic-derived SOA during clean periods? Is it possible that biogenic derived SOA contribute more to SOA formation yielding larger NPF? A better comparison of the MS and PMF data is needed to better understand this aspect. In addition, what is the source of the big particles early morning during the polluted events? How do the meteorological conditions impact NPF?

The reviewer suggested an excellent point. Biogenic derived SOA possibly contributes more to SOA formation yielding larger NPF. Unfortunately, it is very difficult to separate biogenic derived SOA from OOA use quadrupole ACSM in summer. The source of the big particles in early morning during the polluted events is the transport of pollutants from urban sites located to the southeast. We checked the dependence of GR on source region to investigate the meteorological conditions impact NPF, but no clear relationship was found (Figure 11). It showed that new particle growth events were observed in different footprints, suggesting that NPF was not limited by meteorological conditions.

References:

Chen, C., Sun, Y. L., Xu, W. Q., Du, W., Zhou, L. B., Han, T. T., Wang, Q. Q., Fu, P. Q., Wang, Z. F., Gao, Z. Q., Zhang, Q., and Worsnop, D. R.: Characteristics and sources of submicron aerosols above the urban canopy (260 m) in Beijing, China, during the 2014 APEC summit, Atmos. Chem. Phys., 15, 12879-12895, 2015.

DeCarlo, P. F., Dunlea, E. J., Kimmel, J. R., Aiken, A. C., Sueper, D., Crounse, J., Wennberg, P. O., Emmons, L., Shinozuka, Y., Clarke, A., Zhou, J., Tomlinson, J., Collins, D. R., Knapp, D., Weinheimer, A. J., Montzka, D. D., Campos, T., and Jimenez, J. L.: Fast airborne aerosol size and chemistry measurements above Mexico City and Central Mexico during the MILAGRO campaign, Atmos. Chem. Phys., 8, 4027-4048, 10.5194/acp-8-4027-2008, 2008.

de Gouw, J. A., Welsh-Bon, D., Warneke, C., Kuster, W. C., Alexander, L., Baker, A. K., Beyersdorf, A. J., Blake, D. R., Canagaratna, M., Celada, A. T., Huey, L. G., Junkermann, W., Onasch, T. B., Salcido, A., Sjostedt, S. J., Sullivan, A. P., Tanner, D. J., Vargas, O., Weber, R. J., Worsnop, D. R., Yu, X. Y., and Zaveri, R.: Emission and chemistry of organic carbon in the gas and aerosol phase at a sub-urban site near Mexico City in March 2006 during the MILAGRO study, Atmos. Chem. Phys., 9,

3425-3442, 10.5194/acp-9-3425-2009, 2009.

Hu, W. W., Hu, M., Yuan, B., Jimenez, J. L., Tang, Q., Peng, J. F., Hu, W., Shao, M., Wang, M., Zeng, L. M., Wu, Y. S., Gong, Z. H., Huang, X. F., and He, L. Y.: Insights on organic aerosol aging and the influence of coal combustion at a regional receptor site of central eastern China, Atmos. Chem. Phys., 13, 10095-10112, 10.5194/acp-13-10095-2013, 2013.

Hu, W., Hu, M., Hu, W., Jimenez, J. L., Yuan, B., Chen, W., Wang, M., Wu, Y., Chen, C.,Wang, Z., Peng, J., Zeng, L., and Shao, M.: Chemical composition, sources and aging process of submicron aerosols in Beijing: contrast between summer and winter, J. Geophys. Res., 121, 1955-1977, 10.1002/2015JD024020, 2016.

Hu, W., Hu, M., Hu, W.-W., Zheng, J., Chen, C., Wu, Y., and Guo, S.: Seasonal variations in high time-resolved chemical compositions, sources, and evolution of atmospheric submicron aerosols in the megacity Beijing, Atmos. Chem. Phys., 17, 9979-10000, 10.5194/acp-17-9979-2017, 2017.

Huang, X.-F., He, L.-Y., Hu, M., Canagaratna, M., Sun, Y., Zhang, Q., Zhu, T., Xue, L., Zeng, L.-W., and Liu, X.-G.: Highly time-resolved chemical characterization of atmospheric submicron particles during 2008 Beijing Olympic Games using an Aerodyne High-Resolution Aerosol Mass Spectrometer, Atmos. Chem. Phys., 10, 8933-8945, 2010.

Jiang, Q., Sun, Y., Wang, Z. and Yin, Y.: Real-time online measurements of the inorganic and organic composition of haze fine particles with an Aerosol Chemical Speciation Monitor (ACSM), Chin Sci Bull, 2013, 58: 3818-3828. (in Chinese)

Middlebrook, A. M., Bahreini, R., Jimenez, J. L., and Canagaratna, M. R.: Evaluation of composition-dependent collection efficiencies for the aerodyne aerosol mass spectrometer using field data, Aerosol Sci. Tech., 46, 258-271, 2012.

Ng, N. L., Herndon, S. C., Trimborn, A., Canagaratna, M. R., Croteau, P., Onasch, T. B., Sueper, D., Worsnop, D. R., Zhang, Q., and Sun, Y.: An Aerosol Chemical Speciation Monitor (ACSM) for routine monitoring of the composition and mass concentrations of ambient aerosol, Aerosol Sci. Tech., 45, 780-794, 2011.

Pei, B., Cui, H., Liu, H., and Yan, N.: Chemical characteristics of fine particulate matter emitted from commercial cooking, Front. Environ. Sci. Eng., 10, 559-568, 10.1007/s11783-016-0829-y, 2016.

Sun, J., Zhang, Q., Canagaratna, M. R., Zhang, Y., Ng, N. L., Sun, Y., Jayne, J. T., Zhang, X., Zhang, X., and Worsnop, D. R.: Highly time- and size-resolved characterization of submicron aerosol particles in Beijing using an Aerodyne Aerosol Mass Spectrometer, Atmos. Environ., 44, 131-140, 10.1016/j.atmosenv.2009.03.020, 2010.

Sun, Y., Wang, Z., Dong, H., Yang, T., Li, J., Pan, X., Chen, P., and Jayne, J. T.: Characterization of summer organic and inorganic aerosols in Beijing, China with an Aerosol Chemical Speciation Monitor, Atmos. Environ., 51, 250-259, 10.1016/j.atmosenv.2012.01.013, 2012.

Sun, Y., Xu, W., Zhang, Q., Jiang, Q., Canonaco, F., Prévôt, A. S. H., Fu, P., Li, J., Jayne, J., Worsnop, D. R., and Wang, Z.: Source apportionment of organic aerosol from two-year highly time-resolved measurements by an aerosol chemical speciation

monitor in Beijing, China, Atmos. Chem. Phys. Discussion., 1-33, 2018.

Weber, R. J., Guo, H., Russell, A. G., and Nenes, A.: High aerosol acidity despite declining atmospheric sulfate concentrations over the past 15 years, Nature Geoscience, 9, 282-+, 10.1038/ngeo2665, 2016.

Xu, J., Zhang, Q., Chen, M., Ge, X., Ren, J., and Qin, D.: Chemical composition, sources, and processes of urban aerosols during summertime in northwest China: insights from high-resolution aerosol mass spectrometry, Atmos. Chem. Phys., 14, 12593-12611, 10.5194/acp-14-12593-2014, 2014.

Zhang, Y. M., Zhang, X. Y., Sun, J. Y., Lin, W. L., Gong, S. L., Shen, X. J., and Yang, S.: Characterization of new particle and secondary aerosol formation during summertime in Beijing, China, Tellus B, 63, 82-394, 10.1111/j.1600-0889.2011.00533.x, 2011.

Zhang, Y., Tang, L., Croteau, P. L., Favez, O., Sun, Y., Canagaratna, M. R., Wang, Z., Couvidat, F., Albinet, A., Zhang, H., Sciare, J., Prévôt, A. S. H., Jayne, J. T., and Worsnop, D. R.: Field characterization of the PM2.5 Aerosol Chemical Speciation Monitor: insights into the composition, sources, and processes of fine particles in eastern China, Atmos. Chem. Phys., 17, 14501-14517, 10.5194/acp-17-14501-2017, 2017.

Zhao, J., Du, W., Zhang, Y., Wang, Q., Chen, C., Xu, W., Han, T., Wang, Y., Fu, P., Wang, Z., Li, Z., and Sun, Y.: Insights into aerosol chemistry during the 2015 China Victory Day parade: results from simultaneous measurements at ground level and 260 m in Beijing, Atmos. Chem. Phys., 17, 3215-3232, 10.5194/acp-17-3215-2017, 2017.

Zhang, Q., Jimenez, J. L., Canagaratna, M. R., Allan, J. D., Coe, H., Ulbrich, I., Alfarra, M. R., Takami, A., Middlebrook, A. M., Sun, Y. L., Dzepina, K., Dunlea, E., Docherty, K., DeCarlo, P. F., Salcedo, D., Onasch, T., Jayne, J. T., Miyoshi, T., Shimono, A., Hatakeyama, S., Takegawa, N., Kondo, Y., Schneider, J., Drewnick, F., Weimer, S., Demerjian, K., Williams, P., Bower, K., Bahreini, R., Cottrell, L., R.J.Griffin, Rautiainen, J., Sun, J. Y., Zhang, Y. M., and Worsnop, D. R.: Ubiquity and dominance of oxygenated species in organic aerosols in anthropogenically-influenced northern hemisphere mid-latitudes, Geophys. Res. Lett., 34, L13801, doi:10.1029/2007GL029979, 2007.

---

## Author Response (AR2)

Dear Editor Su,

We appreciate the reviewer #2's further comments on our manuscript. Following the reviewer#2's suggestions, we expanded the related discussions (in blue) in the revised manuscript including:

(1) Added more information on CE. "In fact, the differences in mass concentrations were less than 5% between composition-dependent CE and a constant CE of 0.5 in this study."

(2) Added a paragraph to describe the estimations of HOA and COA using BC. "We further evaluated the factors of HOA and COA assuming that BC is dominantly from traffic emissions while the contribution from cooking emissions is minor (Sun et al., 2018). POA from the two-factor solution was highly correlated ($r^2$ > 0.66) with BC between 1:00 – 10:00 when cooking emissions were not significant (Fig. S8). The ratios of POA/BC were also the lowest during this period, suggesting the dominant contribution of HOA to POA. We then used the average ratio of POA/BC (0.62) during this period to estimate the concentrations of HOA and COA. The estimated HOA and COA on average contributed both 11% to OA, consistent with the results of ME-2 analysis. This suggested that the results from ME-2 analysis are reasonable."

(3) Figures R4-R5 were added in supplementary as Figure S17 to support our conclusions.

We also went through the manuscript and make some obvious grammar mistakes.

[Figure]

**Figure S17.** (a) The relationship between GR and CS in the OOA fraction of more than 32% conditions, (b) The relationship between GR and sulfate concentration in the sulfate concentration of less than 3 μg m⁻³ conditions.

We believe that we have addressed the reviewer #2's comments satisfactorily, and we look forward hearing back from you.

Sincerely,

Yele Sun, Ph.D., Professor

[revised manuscript text omitted]

---

## Author Response (AR3)

Response to Editor's comments.

I have one remaining concern about the correlation between GR and CS. The GR depends on amount of condensable vapors, and on the contrary a higher CS will reduce the vapor concentration and GR. Thus the apparent correlation should have other explanations and it could be misleading for the way it is presented and discussed in the current manuscript.
I'd suggest to change the relevant discussions in the text as well as the statement in the abstract.

We appreciate your comments on the correlation between GR and CS. We agree with you that a higher condensation sink is expected to decrease particle growth rates by consuming the condensable vapours faster. The positive correlation between GR and CS might indicate that the source of condensable vapours is closely connected to CS, which can result from the strong contribution of the (semi-)condensable vapours to the build-up of CS prior to the observation. Another possible explanation is heterogeneous surface chemistry is more important than the CS for aerosol particle growth in highly polluted environment (Kulmala et al., 2017).

Following the editor's suggestions, we added the possible explanations in the text, and deleted the strong statements in the abstract and conclusions.

It now reads in the text as:

[revised manuscript text omitted]